# Analysis of GEFS-Aerosols annual budget to better understand the aerosol predictions simulated in the model

Li Pan[1], Partha S. Bhattacharjee[2], Li (Kate) Zhang[3,4], Raffaele Montuoro[5], Barry Baker[6], Jeff McQueen[5], Georg A. Grell[4], Stuart A. McKeen[3,7,*], Shobha Kondragunta[8], Xiaoyang Zhang[9], Gregory J. Frost[7], Fanglin Yang[5], Ivanka Stajner[5]

[1]Lynker Technologies at NCEP/NWS/EMC, Leesburg, VA, US;

[2]SAIC at NCEP/NWS/EMC, College Park. MD, US;

[3]CIRES, University of Colorado, Boulder, CO, US;

[4]Global Systems Laboratory, NOAA, Boulder, CO, US;

[5]Environmental Modeling Center, National Weather Service, Greenbelt, MD, US;

[6]NOAA Air Resources Laboratory, College Park, MD, US;

[7]Chemical Sciences Laboratory, NOAA, Boulder, CO, US;

[8]NOAA/NESDIS Center for Satellite Applications and Research, Greenbelt, MD, US;

[9]Department of Geography, South Dakota State University, Brookings, SD, US

* Retired

*Correspondence to:* Li Pan (Li.Pan@noaa.gov)

**Abstract.** In September 2020, a global aerosol forecasting model was implemented as an ensemble member of the National Oceanic and Atmospheric Administration (NOAA) National Centers for Environmental Prediction (NCEP) Global Ensemble Forecasting System (GEFS) v12.0.1 (hereafter referred to as "GEFS-Aerosols"). In this study, GEFS-Aerosols simulation results from September 1, 2019 to September 30, 2020 were evaluated using an aerosol budget analysis. These results were compared with results from other global models as well as reanalysis data. From this analysis, the global average lifetimes of black carbon (BC), organic carbon (OC), dust, sea salt, and sulfate are 4.06, 4.29, 4.59, 0.34 and 3.3 days, respectively, with the annual average loads of 0.135, 1.29, 4.52, 6.80 and 0.50 TG. Compared to National Aeronautics and Space Administration (NASA)'s Goddard Earth Observing System-Goddard Chemistry Aerosol and Radiation Transport Model (GEOS4-GOCART), the aerosols in GEFS-Aerosols have a relatively short lifetime because of the faster removal processes in GEFS-Aerosols. Meanwhile, in GEFS-Aerosols, aerosol emissions are the determining factor for the mass and composition of aerosols in the atmosphere. The size (bin) distribution of aerosol emissions is as important as its total emissions, especially in simulations of dust and sea salt. Also most importantly, the strong monthly and interannual variations in natural sources of

aerosols in GEFS-Aerosols suggests that improving the accuracy of prognostic concentrations of aerosols is important for

applying aerosol feedback to weather and climate predictions.

## 1. Introduction

Aerosol direct and indirect radiative effects greatly affect weather and climate [Ramanathan et al., 2001; Haywood and Boucher, 2000; Kaufman et al., 2005; Satheesh and Moorthy, 2005]. Since the Industrial Revolution, human activities have dramatically altered the composition and distribution of aerosols not only on a regional but also on a global scale [Tsigaridis

et al., 2006; Naik et al., 2013]. To accurately predict weather and climate, it is increasingly important to correctly characterize aerosol behavior and its evolution in the atmosphere [Takemura et al., 2005; Yukimoto et al., 2012]. As a first step towards this goal, NCEP implemented the latest version of its global aerosol forecasting model in the Global Ensemble Forecasting System (GEFS) [Hamill et al., 2013] in September 2020 to replace the decommissioned NCEP Environmental Modeling System (NEMS) GFS Aerosol Component (NGAC) [Bhattacharjee et al., 2018; Wang et al., 2018].

Bhattacharjee et al. [2023] evaluated the simulation results of the GEFS-Aerosols model using AOD (aerosol optical depth) data derived from satellite retrieval (MODIS and VIIRS), AOD data simulated by other models (MEERA2 and NGAC), and AOD data observed from 50 AERONET stations. The period of evaluation from August 2019 to August 2020 almost coincides with the time period of this study, namely from September 2019 to September 2020. In addition to the regular daily or monthly forecast evaluations of GEFS-Aerosols, three special events were also utilized to evaluate the performance of GEFS-Aerosols.

These include dust events in Northwest Africa, agricultural fires in northern India and the August fire complex in northern California. Zhang et al., [2022b] evaluated not only the AOD simulated by GEFS-Aerosols from 5 July to 30 November 2019, but also the aerosol concentrations simulated by GEFS-Aerosols during the 22-month ATOM (Atmospheric TOmography Mission) period from 2016 to 2019.

These assessments found that GEFS-Aerosols captures not only major wildfire plumes in southern Africa, Siberia, the central

Amazon, and central South America, as well as agricultural fire plumes over India, but also high dust events in North Africa and the Arabian Peninsula; At the same time, GEFS-Aerosols has good performance in reproducing the seasonal variations at most surface observation sites dominated by dust and biomass plumes, as well as reproducing the vertical profiles of OC, BC, sulfate, dust and sea salt observed by ATOM. However, these findings are based on comparisons of AOD or aerosol concentrations and lack other assessments beyond AOD and concentration.

In this paper, instead of focusing on aerosol concentration and AOD in a general aerosol evaluation, an aerosol budget analysis [Hodzic et al., 2016] was used to evaluate GEFS-Aerosols. This method relies on the aerosol mass balance equation in the chemical transport model, which enables a budget assessment of aerosol chemical and physical processes (such as emissions, deposition and reactions) because these processes are the determinants of aerosol concentration. This article is structured as follows: first, an aerosol mass balance equation is introduced to check the aerosol emission and deposition values calculated

in the analysis; then, their contribution to the ambient aerosol concentration is estimated; finally, the annual budget for aerosols

in GEFS-Aerosols is analyzed and compared with results from other global aerosol models such as NASA's GEOS4-GOCART [Colarco et al., 2010].

## 2. Methodology

### 2.1. GEFS-Aerosols

GEFS-Aerosols is an inline chemical prediction model that is fully coupled to the NCEP Global Prediction System FV3GFS (GFS v15 with FV3 Dynamic Core) [Putman and Lin, 2007] using the Earth System Modeling Framework (ESMF) based National Joint Operational Prediction Capability (NUOPC) layer [Zhang et al., 2022b]. The chemical mechanism of GEFS-Aerosols is built upon the Weather Research and Forecasting module coupled with chemistry (WRF-CHEM) and NASA-GOCART, and was developed by NASA Goddard Space Flight Center (GSFC) in conjunction with NOAA Global Systems

Laboratory (GSL) and Chemical Science Laboratory (CSL) [Zhang et al., 2022a]. The model predicts 5 aerosols for 120 hours at a grid resolution of 0.25 x 0.25 degrees, with four forecast cycles per day: 00z, 06z, 12z and 18z. The output can be downloaded online at (https://nomads.ncep.noaa.gov/pub/data/nccf/com/gens/prod).

In GEFS-Aerosols, both the BC and OC are divided into two bins: hydrophobic (BC1 and OC1) and hydrophilic (BC2 and OC2). Dust comes in five size bins: 0.1-1.0, 1.0-1.8, 1.8-3.0, 3.0-6.0, and 6.0-10.0 microns, referred to in the discussion as

DUST1, DUST2, DUST3, DUST4, and DUST5. Sea salt also has five size bins: 0.03-0.1, 0.1-0.5, 0.5-1.5, 1.5-5.0, and 5.0-10.0 microns, called SS1, SS2, SS3, SS4, and SS5, respectively. There is only one bin for sulfate.

Aerosols are removed from the atmosphere by wet and dry deposition as well as by gravitational settling in GEFS-Aerosols. Wet deposition in GEFS-Aerosols is the sum of large-scale wet removals and convective scavenging. The large-scale wet removal scheme is from WRF-CHEM (https://github.com/wrf-model/WRF/blob/master/chem/module_wetdep_ls.F), and the

convective scavenge is calculated in FV3GFS physics and is based on the simplified Arakawa-Schubert (SAS) scheme [Pan et al, 1994; Zhang et al., 2022b]. The computation of gravitational settling for dust and sea salt are based on the updated finite-difference scheme in WRF-CHEM [Ukhov et al., 2021]. This scheme not only calculates the gravity settlement from the upper layer of the model to the lower layer of the model, but also calculates the gravity settlement from the bottom layer of the model to the ground. Meanwhile, the GOCART dry deposition protocol [Chin et al., 2000] was used for GEFS-Aerosols. Dry

deposition in GEFS-aerosol was calculated by dry deposition velocity based on aerodynamic resistance, sublayer resistance and surface resistance. Therefore, aerosol gravity deposition and aerosol dry deposition are completely separated in GEFS-Aerosols.

In this analysis, the time period of retrospective simulation is from September 1, 2019 to September 30, 2020. This retrospective simulation was restarted every day at 00z with updated meteorological initial conditions and conducted for the

24-hour free forecast duration. The simulation has a global horizontal resolution of 0.25 x 0.25 degrees and 64 vertical layers which are based on sigma-p levels from the surface to 200 Pa. Anthropogenic emissions of BC, OC and $SO_2$ are from the Community Emissions Data System (CEDS) 2014 edition [Hoesly et al., 2018]. The Blended Global Biomass Combustion

Emissions Product Version 3 (GBBEPx-v3) provides daily updated biomass burning emissions data for BC, OC and SO$_2$ as a function of Fire Radiative Power (FRP) [Zhang et al., 2011]. The wind blown dust scheme used in GEFS-Aerosols is FENGSHA [Dong et al., 2016; Zhang et al., 2022b], and the sea salt scheme is based on NASA's second-generation GOCART model [Colarco et al., 2010].

## 2.2. Mass Balance Equation

Aerosol processes should obey the law of conservation of mass. For a given system (such as the entire atmosphere), the amount of chemicals entering the system is equal to the amount of chemicals leaving the system. In the global model (GEFS-Aerosols), the law of conservation of mass can be simply expressed as the mass balance equation:

Initial + Emissions + Reactions = Final + Removal                                                                (Eq. 1)

in which, Initial and Final are the mass of aerosols at the beginning and end of the model simulation, respectively; emissions are the total amount of aerosols emitted by anthropogenic and natural sources in the model simulation; reactions are the production and loss due to chemical reactions; removal is the cumulative removal of aerosol mass from the atmosphere due to wet and dry deposition and gravitational settling. Although aerosol mass is affected by advection, diffusion, and physical processes, these processes are not specifically considered in the equations because they do not cause aerosols to leave/enter the system or change the aerosol species. However, these processes do change the concentration of aerosols in the atmosphere, and this effect is ultimately included in the "Initial", "Reaction", "Removal" and "Final" terms of the mass balance equation.

## 2.3. Model Error and Initial Error

During a GEFS-Aerosols simulation, the total mass non-conservation caused by each aerosol chemical and physical process is defined as the model error and is calculated as:

$$\text{Model}_{err} = \frac{\text{ABS(Initial+ Emissions+Reactions} - \text{Final \ \ Removal)}}{\text{Initial}} \times 100\% \qquad \text{(Eq. 2)}$$

Treatment of initial conditions in the model may also lead to non-conservation of aerosol mass in the GEFS-Aerosols simulation if the model is run as a cycled simulation with meteorological inputs updated every 24 hours. Fig. 1 shows a flowchart of the GEFS-aerosol forecasting workflow. To implement GEFS-Aerosols forecasts, the model needs to read the aerosol restart file from the end of the last integration time period, GFS initial conditions, and meteorological incremental data defined below. The GFS initial conditions generated from FV3GFS or FV3GFS-GDAS (Global Data Assimilation System) [Kleist et al., 2009] analysis files are used to update the meteorological initial conditions in GEFS-Aerosols to reduce the bias of the meteorological initial input. The major advantage of this initial process is to keep the aerosol input field unchanged, but maximize the use of the assimilated meteorological field. The disadvantage of this approach is that it results in a non-conservation of mass in GEFS-Aerosols as the air density has changed but the aerosol mixing ratio remains the same as before. In awareness of this problem, meteorological incremental data (defined as the difference between the GFS initial conditions and the meteorological predictions from the previous GEFS-aerosols simulation cycle) are used to reduce this inconsistency.

This error caused by the mismatch between aerosol initial conditions and meteorological initial conditions is defined as the initial error and is calculated as:

$$Initial_{err} = \frac{Mass\_pre - Mass\_cur}{Mass\_pre} x 100\% \qquad \text{(Eq. 3)}$$

in which, $Mass\_pre$ represents the aerosol mass at the last time step of the previous cycle and $Mass\_cur$ represents the aerosol mass at the first time step of the current cycle.

## 3. Results and Discussions

### 3.1. Budget analysis

GEFS-aerosols was evaluated in Zhang et al. [2022b] and Bhattacharjee et al. [2023] by comparing model-simulated aerosol concentrations and derived Aerosol Optical Depth (AOD) values with in-situ observations, satellite retrievals, and aircraft measurements during the ATOM-1campaign [Brock et al., 2019]. In this study, GEFS-Aerosols is further evaluated using an aerosol mass budget analysis. This method allows one to examine major chemical and physical processes, such as emissions, removals, and reactions. These processes ultimately determined 3-d aerosol distribution, which in term affect concentration and AOD. For example, Fig. 2 represents the monthly mean sea salt AOD (top) and surface mass concentration ($\mu g/m^3$) (bottom) simulated in October 2019 from GEFS-Aerosols (left) and Modern-Era Retrospective analysis for Research and Applications Version 2 (MERRA-2) (right) [Molod et al., 2015]. GEFS-Aerosols and MERRA-2 show very similar results in simulating sea salt AOD, as does the distribution pattern of sea salt surface mass concentration. This is due to the fact that in GEFS-Aerosols, AOD is calculated using look-up tables (LUTs) of aerosol optical properties in the NASA GOCART model, consistent with AOD calculations in MERRA2. However, comparisons of sea salt emission, sedimentation, and wet and dry deposition, in Figs. 3 and 4, show large discrepancies in those processes between GEFS-Aerosols and MERRA-2. For example, sea salt emissions in GEFS-Aerosols [Zhang et al., 2022b] are approximately 3 times higher than in MERRA-2 [Randles et al., 2017]; dry deposition of sea salt in GEFS-Aerosols is also at least twice as high as in MERRA-2; and there are also significant differences in wet deposition and sedimentation. This is a good example of how two models can have completely different sources and sinks but end up with very similar concentration predictions, suggesting the importance of evaluating aerosol physical and chemical processes in the GEFS-Aerosols model.

### 3.2. Mass Conservation

Fig. 5 shows the BC and OC initial errors (%) at the beginning of each month for the GEFS-Aerosols simulation time period from October 2019 to October 2020. The mean initial errors for BC and OC are 0.01% and 0.017%, respectively. The magnitude of the initial error depends on the degree to which the meteorological fields of the restart file and the reanalysis file differ and how much this difference overlaps with the spatial distribution of aerosols; therefore, OC, as a type of aerosol with a mass greater than BC in GEFS-Aerosols, has an initial error that is usually greater than that of BC. Note that the initial error

can be positive or negative, appearing randomly. For example, the initial error of BC is positive in October 2019 but negative in October 2020, which means that GEFS-Aerosols has no systematic bias in handling the initial conditions, and the errors could cancel each other out in successive runs of the model. Therefore, the cumulative initial error of the model is negligible; otherwise even BC, with the smallest initial error of the different types in GEFS-Aerosols, could result in a 3.7% (0.01% per day x 365 days) mass non-conservation error as the model was restarted every day in GEFS-Aerosols simulations through a year. In general, the absolute value of the initial error is less than 0.13% for any aerosol type in GEFS-Aerosols. Without "incremental" adjustments to the meteorological initial conditions (Fig. 1), tests have shown that the initial error can be as high as 10% each time the model is restarted. This analysis suggests that the way GEFS-Aerosols handles initial conditions is not perfect but is acceptable.

Fig. 6 shows the absolute value of model error (% -> brown line, Primary Axis) for BC, OC, dust, and sea salt for each day during January 2020. Model error for sulfate was not assessed in this study. Sulfate is the product of the oxidation reaction of DMS (dimethyl sulfide), MSA (methanesulfonic acid), and $SO_2$, the current GEFS-Aerosols diagnostic system does not support outputting the amount of sulfate produced by these reactions. In January 2020, the maximum model errors for BC, OC, dust, and sea salt per hour were less than 0.8%, 1.2%, 3%, and 0.45%, respectively. The larger model errors for dust, OC and BC coincide with emission outbreaks for dust, OC and BC, which are shown as the emission changes (kg/s -> blue line, Secondary Axis) in Fig. 6. In terms of annual averages of absolute values, the model errors are 0.04%, 0.05%, 0.35% and 0.08% for BC, OC, dust and sea salt, respectively. The model errors for dust are larger than those for BC, OC, and sea salt.

Theoretically, aerosol mass in GEFS-Aerosols simulations should be conserved, which means that the model error should be zero if calculation accuracy is not taken into account. Possible reasons for the non-conservation of aerosol mass in GEFS-Aerosols as shown in Fig. 6 include: 1) The aerosol mass is not conserved in the advection, diffusion and physical processes of the model; 2) Aerosol leakage at the top of the model layer; 3) There are problems with calculating aerosol terms (such as emissions and deposition) in the mass balance equations;

First, aerosol transport in GEFS-Aerosols is based on the FV3 dynamic core [Lin et al., 1994], which is also used in NASA-GOCART and GEOS-Chem. The mass conservation problem of this dynamical framework has been discussed by Lin and Rood [1996]. The physical processes of GEFS-aerosols are derived from the GFDL (Geophysical Fluid Dynamics Laboratory) cloud microphysics scheme [Lin et al., 1983], which strictly adheres to the conservation of moist energy during phase changes. Secondly, the pressure at the top of the model in GEFS-Aerosols is set to 200 Pa. Since the pressure at the top of the model is low enough and the layers of the model are dense enough near the top [Campbell et al., 2022], the aerosol concentration in GEFS-Aerosols is the background concentration ($1 \times 10^{-16}$ µg/kg) in these layers. There may be mass conservation issues at the top of the model, but their impact is minimal. To better understand the model error shown in Fig. 6, the GEFS-Aerosols output frequency was changed from every 3 hours (orange line in Fig. 7) to every hour (blue line in Fig. 7). Fig. 7 shows that the hourly variation in model error for dust simulations (in a 5-day simulation) is actually similar to that at 3-hour intervals, but the magnitude of peaks is reduced by about 60%, suggesting that model error is sensitive to model output frequency.

The linear assumption of aerosol deposition and emissions when testing the mass balance equation (Eq. 1) are the main cause of the model error. The deposition and emissions output by the GEFS-Aerosols diagnostic system are instantaneous values rather than cumulative values. Therefore, to calculate the cumulative amount of aerosol deposition or emissions over a model output time interval (e.g., three hours), simply multiply this value by three based on the linearity assumption. This treatment only affects deposition calculations for BC and OC (The daily emissions of BC and OC are constant.), but for dust and sea salt it affects not only deposition but also emissions calculations. At the same time, the wind threshold velocity makes the dust emissions more nonlinear than the source or sink terms of the other aerosol types. Therefore, the model errors for dust and sea salt are higher than those for BC and OC, while the model errors for dust are the highest. In general, when aerosol deposition or emissions increase, the error in calculating them in the analysis also increases due to linearity assumptions. For example, when aerosol emissions increase, in addition to BC and OC, the error in calculating aerosol emissions in the mass balance equation also increases. Correspondingly, the error in deposition calculations will also increase because an increase in ambient aerosol concentration will lead to an increase in the amount of deposition. This is why the model error for dust has a higher correlation with aerosol emissions than for BC and OC. However, for sea salt this correlation does not exist because sea salt emissions shown in Figure 6 are relatively stable.

Because the linearity assumption in the analysis can lead to model errors as mentioned above, the model error shown in Fig. 6 does not represent the true model simulation error, but rather the calculation error in the mass conservation analysis. Therefore the main purpose of using the mass balance equation in this study is to verify the aerosol deposition and emissions calculated in the model budget analysis, rather than verifying whether aerosol mass is conserved in the GEFS-Aerosols. Like initial errors, model errors cancel each other out in the full-year analysis and do not affect our conclusions. For the full year, the total model errors (%) for BC, OC, Dust and Sea Salt are -0.0206, -0.0218, 0.00038 and -0.0333, respectively.

### 3.3. Global Aerosol Masses

Fig. 8 shows the changes in the total aerosol mass of BC, OC, dust, sea salt and sulfate in the atmosphere from September 1, 2019 to September 1, 2020. For BC (Fig.. 8a), the annual mean BC mass in the atmosphere is $1.35 \times 10^8$ kg, with a maximum of $2.08 \times 10^8$ kg on January, 2020 and a minimum of $9.45 \times 10^7$ kg on June, 2020. For OC (Fig. 8b), the annual mean OC mass in the atmosphere is $1.09 \times 10^9$ kg. The maximum and minimum values are $2.18 \times 10^9$ kg and $7.75 \times 10^8$ kg, respectively, which appeared on the same day as BC. For dust (Fig. 8c), the annual mean mass is $4.52 \times 10^9$ kg, with maximum and minimum values of $9.02 \times 10^9$ kg and $1.31 \times 10^9$ kg on June, 2019 and November, 2019, respectively. For sea salt (Fig. 8d), the annual mean sea salt mass in the atmosphere is $6.80 \times 10^9$ kg. The maximum value of sea salt mass is $8.19 \times 10^9$ kg, which occurred on May, 2020, and the minimum value of sea salt mass is $5.59 \times 10^9$ kg, which occurred on October, 2019. In term of sulfate (Fig. 8e), the annual mean mass in the atmosphere is $5.04 \times 10^8$ kg. The maximum value is $7.11 \times 10^8$ kg, which occurred on September, 2019, and the minimum value is $3.44 \times 10^8$ kg, which occurred on January, 2020.

In GEFS-Aerosols, sea salt is the most dominant aerosol, followed by dust, OC, sulfate, and BC, where OC was about 10 times the mass of BC. In the simulated year, the trends for BC and OC masses are decreasing (16.4% and 22.3%, respectively) and

the trends for dust and sea salt are increasing (24.9% and 16.0%, respectively); for sulfate the trend is almost constant with only a very slight decrease (8.09%).

### 3.4. Emissions (Table. 1)

The emissions of BC come from anthropogenic sources and biomass burning, of which anthropogenic emissions account for about 66.9% of the total emissions, with the maximum and minimum values being 84.5% and 30.2%, respectively. The emissions of OC are similar to that of BC in GEFS-Aerosols. On average, OC anthropogenic emissions account for 50% of OC emissions, and this contribution can be as large as 73.5% and as small as 13.4%. Biomass burning contributes more to OC than to BC. 100% of BC and OC emissions are assumed to be hydrophobic (BC1 and OC1) and will typically transition to hydrophilic (BC2 and OC2) in the atmosphere within two and a half days [Maria et al., 2004].

In GEFS-Aerosols, more than 60% of dust emissions are coarse particles typically larger than 2.5 μm in diameter, 10.1% of dust emissions are in DUST1 (0.1-1.0 μm), 10.1% of dust emissions are in DUST2 (1.0-1.8 μm), 20.9% of dust emissions are in DUST3 (1.8-3.0 μm), 48.5% of dust emissions are in DUST4 (3.0-6.0 μm), and 10.3% of dust emissions are in DUST5 (6.0-10.0 μm). For sea salt, 49.8% of sea salt emissions come from SS4 (1.5-5.0 μm), 37.4% of sea salt emissions from SS5 (5.0-10.0 μm), 11.7% of sea salt emissions from SS3 (0.5-1.5 μm), 1.1% of sea salt emissions from SS2 (0.1-0.5 μm) and 0.034% of sea salt emissions from SS1 (0.03-0.1 μm). There are no direct sulfate emissions in GEFS-Aerosols. $SO_2$ and DMS emissions are sources of sulfate via atmospheric reactions. $SO_2$ comes from anthropogenic sources and biomass burning, with anthropogenic emissions contributing more than 99%. DMS is emitted from the ocean.

Aerosol emissions are directly and indirectly related to their mass in the atmosphere. For BC, OC, dust, and sea salt, which are emitted directly from their respective sources, Fig. 8a shows that BC mass is highly correlated with its emissions, with a linear regression coefficient (R) of 0.64. OC mass is also highly correlated with OC emissions, with an R value of 0.58 (Fig. 8b). The correlation coefficient between sea salt mass and emissions is 0.89, the highest for any aerosol type in GEFS-Aerosols (Fig. 8d). For dust, when dust emissions peak, the dust mass increases accordingly. However, the correlation coefficient between dust mass and dust emissions R is only around 0.33 (Fig. 8c). For sulfate, anthropogenic $SO_2$ emissions are indirectly related to ambient sulfate concentrations through the $SO_2$ oxidation reaction.

### 3.5. Removals

As shown in Table 2, dry deposition contributes about 25% of the BC removed from the atmosphere; 72% of dry deposition comes from hydrophobic BC and 28% from hydrophilic BC (Table. 1). Similarly, 25% of OC is removed by dry deposition, of which hydrophobic OC accounts for 72% and hydrophilic OC accounts for 28% (Table. 1). On the other hand, dry deposition accounts for about 17.3% of the total atmospheric dust removal. DUST4 and DUST3 contribute 40.9% and 25.9%, respectively, to dust dry deposition, much larger than DUST1 (14.9%), DUST2 (14.2%) and DUST5 (4.1%) (Table. 1). For sea salt removal, dry deposition contributes only 4.3% (Table. 2), of which SS4, SS5 and SS3 contribute 61.1%, 19.7% and

17.4%, respectively (Table. 1). For sulfate, dry deposition accounts for approximately 9.5% (Table. 2) of the total sulfate removal.

As also shown in Table 2, wet deposition contributes about 75% of the total BC removed from the atmosphere, of which 43% comes from hydrophobic BC and 57% from hydrophilic BC (Table. 1). 75% of the OC is removed due to wet deposition, of which 41% are hydrophobic and 59% are hydrophilic (Table. 1). The contribution of dust wet deposition to dust removal is about 36.5% (Table. 2), of which DUST4 accounts for 29.5%, followed by DUST3 with 29.4%, DUST1 with 20.9%, DUST2 with 18.7%, and DUST5 with 1.4% (Table. 1). For sea salt removal, 59% (Table. 2) of the contribution comes from wet deposition, with SS4 contributing the most at 62.2%, SS5 accounting for 18.3%, and SS3 accounting for 17.7% (Table. 1). Wet deposition contributes 90.5% of sulfate removal (Table. 2).

Gravitational settling is only applied to dust and sea salt in GEFS-Aerosols. Gravity sedimentation contributes 46.3% of the total dust removal, which is greater than wet deposition (36.5%) and dry deposition (17.3%) terms (Table. 2). The dust coarse mode that mainly includes DUST4 and DUST5 contributes more than 84% (Table. 1) of dust's sedimentation. Gravitational sedimentation accounts for 36.7% of the total removal of sea salt, which is smaller than wet deposition (59%) and larger than dry deposition (4.3%) terms (Table. 2). SS5 has the most sedimentation at about 70.1%, and together with SS4, coarse model sea salt sedimentation exceeds 98.6% (Table. 1) of the total.

In general, these results indicate that both wet and dry deposition and gravitational settling are important for aerosol removal processes in the atmosphere, and the total removal of aerosols (i.e. the sum of wet, dry and sedimentation) is almost equal to their total emissions (Table. 2). However, aerosol dry deposition is typically not the dominant process. In addition, the contribution of each aerosol size bin to its total removal is close to its emission distribution. For example, dust emissions are divided into five bins, accounting for 10.2%, 10.3%, 20.9%, 48.3% and 10.1% of the total emissions. The corresponding contributions of the five bins to the total dust removal are 10.3%, 10.3%, 21.0%, 48.3% and 10.1% (Table. 1). This result suggests that no process in the GEFS-Aerosols simulations alter the size of the dust particles, so dust is removed in the same proportion as it was emitted. Furthermore, the size distribution of aerosol emissions becomes too important for the removal process in GEFS-Aerosols simulations when the aerosol particle size is not changed in the model. On the other hand, however, OC and BC do not follow a similar rule, as hydrophilic and hydrophobic contributions to their total removal are almost equal, while both types of aerosols are emitted as 100% hydrophobic (Table. 1). OC and BC transition from hydrophobicity to hydrophilicity after emissions, increasing the percentage of hydrophilic species in the atmosphere. Therefore, OC and BC not modeled like dust and sea salt in GEFS-Aerosols, as they do not undergo a size (bin) change.

### 3.6. Reactions

In GEFS-Aerosols, sulfate is the product of the oxidation of $SO_2$ with OH (in the gaseous phase) and $H_2O_2$ (in the aqueous phase):

$SO_2 + OH (H_2O_2) ->$ sulfate                                                                                                (Re. 1)

in which SO$_2$ is emitted from anthropogenic sources and biomass burning, and from the reaction of DMS with OH, H$_2$O$_2$, and NO$_3$.

DMS + OH (H$_2$O$_2$; NO$_3$) -> MSA + SO$_2$;                                (Re. 2)

The oxidant concentrations (OH, H$_2$O$_2$ and NO$_3$) used in the GEFS-Aerosols are prescribed concentrations in the model. The amount of sulfate produced by SO$_2$ oxidation was calculated by assuming that sulfate mass is conserved in Equation 1. Due to seasonal changes in anthropogenic SO$_2$ emissions, more sulfate is produced in December and January than in June and July (Fig. 9). In a one-year GEFS-Aerosols simulation, the sulfate mass decreases by 4.72 x 10$^7$ kg (8.09%) from 5.83 x 10$^8$ kg on

September 1, 2019, to 5.36 x 10$^8$ kg on September 1, 2020 (Fig. 8e). This shows that more sulfate was removed from the atmosphere than was produced during this period. Globally, SO$_2$ oxidation produced an average of 6.5 x10$^6$ kg of sulfate per hour, which is about 1.33% of the total mass of sulfate in the atmosphere.

### 3.7. Lifetime

The lifetime ($\tau$) of each aerosol types in the atmosphere is calculated as:

$\tau = \frac{Aerosol\_Mass}{Total\_Removal}$                                                  (Eq. 4)

Due to the different emissions and removal efficiencies of each size (bin) of aerosols, the lifetimes of different aerosols types and sizes are different. Aerosol lifetime affects the aerosol composition in the atmosphere. The lifetime of BC in GEFS-Aerosols for this time period is 4.06 days; hydrophobic BC accounts for 30.3% of the total atmospheric BC and hydrophilic BC accounts for 69.7% of the total atmospheric BC (Table. 1). For OC, its lifetime in the atmosphere is 4.29 days; hydrophobic

and hydrophilic components account for 30.3% and 69.7% of the total OC in the atmosphere, respectively (Table. 1). Dust has five size bins: DUST1 accounts for 20.3% and its lifetime is 6.89 days; DUST2 accounts for 18.1% and its lifetime is 6.16 days; DUST3 accounts for 28.8% and its lifetime is 4.77 days; DUST4 accounts for 31.1% and its lifetime is 2.26 days; DUST5 accounted for 1.7% and its lifetime is 0.61 days (Table. 1). For the five size bins of sea salt aerosols, the lifetimes of SS1, SS2, SS3, SS4 and SS5 are 0.53 days, 0.52 days, 0.48 days, 0.34 days and 0.09 days, respectively. Sea salt is composed of 0.07%

SS1, 2.2% SS2, 21.3% SS3, 63.7% SS4 and 12.7% SS5 (Table. 1). The lifetime of sulfate is 3.3 days (Table. 2).

To compare the aerosol lifetime simulated in GEFS-Aerosols with the results of the NASA GEOS4-GOCART online simulation [Colarco et al., 2010], the aerosol-weighted lifetimes for dust and sea salt were calculated using Equation 5

$\bar{\tau} = \sum_{n=1}^{bin}(f_n\tau_n)$                                                      (Eq. 5)

where f$_n$ is the aerosol composition fraction for each size (bin). The weighted lifetimes for dust and sea salt are 4.59 and 0.34

days, respectively, while the corresponding aerosol lifetimes in GEOS4-GOCART are 5.85 and 0.88 days, respectively (Table. 2). The lifetime of all aerosols types in GEFS-Aerosols are shorter than those in GEOS4-GOCART. The GEFS-Aerosols simulations were performed at 0.25° x 0.25° degrees horizontal resolution and 64 levels of vertical resolution over a one-year period from September 2019 to September 2020. However, the GEOS4-GOCART results are the overall average from 2000 to 2006, with a spatial resolution of 1.25° longitude x 1.0° latitude, and a vertical resolution of 55 vertical layers. By ruling

out these spatial and temporal differences as possible reasons for the discrepancy in the GEFS-Aerosols and GEOS4-GOCART results, the shorter lifetimes of aerosols in GEFS-Aerosols generally indicate less aerosol emission, higher aerosol removal efficiency, or both, in comparison with GEOS4-GOCART. This issue will be revisited in a later section.

### 3.8. Vertical Profiles

Fig. 10 (September, 2019) shows the vertical distribution of aerosol mass percent along the pressure (y-axis) for the 64 model
layers in GEFS-Aerosols. The mass percent is defined as the ratio of the aerosol mass in each model layer to the total column aerosol mass in the grid. Mass percent and pressure values are global averages. Two distinct types of aerosol vertical distributions are shown in Fig. 10.  For sea salt, most of the sea salt accumulates below 800 hPa. The closer to the surface, the more sea salt mass.  For aerosol species other than sea salt, the aerosol mass peaks at pressure levels between 800 and 600 hPa. Above 400 hPa, the mass percent of most aerosols is less than 1.0%. But for sulfate, hydrophilic BC (BC2) and hydrophilic
OC (OC2), the aerosol mass loadings above 400 hPa are still considerable. This behavior is due to the oxidation reaction of $SO_2$ and the hydrophobic to hydrophilic conversion reaction of BC and OC, which occur throughout the depth of the atmosphere, and the lack of corresponding gravitational sedimentation for BC, OC and sulfate. In general, the vertical distribution of aerosols is closely related to the lifetime of the aerosol in the atmosphere: the longer its lifetime, the more likely the aerosol is to be lifted above the ground and transported over long distances.

Schwarz et al., [2010, 2013] observed BC vertical profiles over the Pacific in the High-performance Instrumented Airborne Platform for Environmental Research Pole-to-Pole Observations (HIPPO) campaign. They found that higher BC mass mixing ratios appeared in the middle and upper troposphere, while lower BC mass mixing ratios were uniformly seen in the lower stratosphere. The vertical profiles of aerosol extinction coefficient derived from Cloud-Aerosol Lidar with Orthogonal Polarization (CALIOP) satellite instrument also showed that elevated aerosol extinction was obtained at altitudes between 1.0
km and 2.0 km above the sea level, which varied by region and season [Koffi et al., 2012; Winker et al., 2013]. Both sets of observations are consistent with the GEFS-Aerosols simulations of aerosol vertical distributions. Note that the HIPPO experiment and CALIOP observations are only used for qualitative comparisons in this study because they are of different timing than the GEFS-aerosol simulations.

Since more than 70% of the Earth's surface is covered by ocean and Fig. 10 represents the global average, the vertical profile
of aerosol mass shown in Fig. 10 is more representative of remote areas. This is the main reason why the vertical distribution of aerosols shown in Fig. 10 is consistent with the HIPPO and CALIOP observations. The zonal distribution of aerosols as a function of pressure and latitude is used to reflect the influence of aerosol sources on the vertical distribution of aerosol concentrations [Chin et al., 2000; Textor et al., 2006; Wang et al., 2014]. Fig. 11 shows the zonal and monthly mean (September, 2019) simulated total aerosol mixing ratio (µg/kg_air). With the exception of the Antarctic and Arctic, the
contribution of aerosol surface emissions to aerosol concentrations decreases with increasing altitude, especially at 15N to 30N, where dust from the Sahara dominates. Aerosols can be elevated up to 600 hPa or even 500 hPa, which means that aerosol effects are global, as a result of the multi-day lifetime of most aerosol types and their corresponding long range transport.

The decrease in aerosol concentration with increasing altitude shown in Fig. 11 is significantly different from that shown in Fig. 10. Fig. 10 shows that certain aerosols (e.g. BC, OC, sulfate and dust) are more concentrated at higher altitudes. However, Fig.10 and Fig. 11 are not contradictory. Because Fig. 11 is more representative of the vertical distribution of aerosols near the source (such as over land), while Fig. 10 is more representative of the vertical distribution of aerosols far away from the source (such as over the ocean). Therefore, in the validation of Fig. 10, the HIPPO experiment and CALIOP observations were used because they measured the vertical profile of aerosols in remote areas.

### 3.9. Composition

Aerosols with different particle sizes in the same type of aerosol have different optical properties for absorbing and scattering sunlight, so it is very important to predict the proportion of aerosols with different particle sizes in the total mass of aerosols. Aerosol emissions and aerosol removal are two factors that determine the aerosol composition in the atmosphere in GEFS-Aerosols. Among them, aerosol emissions determine the total amount of each type of aerosol emitted into the atmosphere, and aerosol removal determines the rate at which this type of aerosol is removed from the atmosphere. Table 1 shows the atmospheric composition of each type of aerosol in the GEFS-Aerosols one-year simulation. For dust, since the same dry velocity and wet scavenge factor are applied to all sizes of dust particles, the dust removal rate is only related to gravitational settling and therefore only to its particle size. The coarse mode dust has a shorter lifetime than the fine mode dust. The composition of the coarse mode dust is smaller than its emission percentage, while the composition of the fine dust mode is larger than its emission percentage. As shown in Table 1, DUST1 emits 10.20% but accounts for 20.29% in the atmosphere; DUST4 emits 48.46% but accounts for 31.06% in the atmosphere. The percentage of different particle sizes in emissions is the main factor determining for the aerosol composition in the atmosphere.

### 3.10. Monthly or Interannual Variations

Fig. 12 shows the normalized percent (%) monthly variations of BC, OC, dust, sea salt, and SO2 emissions in GEFS-Aerosols from September 2019 to December 2020. A negative value indicates that the monthly emissions are below average, and a positive value indicates that the monthly emissions are above average. The larger the difference between positive and negative values, the greater the monthly change in this type of aerosol emissions, and the greater the change in the mass of this type of aerosol. The monthly variations in sea salt emissions shown in the Fig. 12 range from -8.9% in October 2019 to 5.9% in January 2020, therefore, no significant monthly or seasonal variations in sea salt aerosol masses were observed in the model annual simulations (Fig.. 8d). Meanwhile, the mass trend for sulfate is the least (8% decrease) among all types of aerosols in GEFS-Aerosols since it is mainly related to anthropogenic emissions of $SO_2$, which did not change much during the 1-year period considered here (Figs. 8e & 9). The monthly variations in $SO_2$ emissions range from a minimum of -6.8% in September 2019 to a maximum of 10.4% in January 2020 (Fig. 12).

On the other hand, BC, OC and dust emissions have large monthly variations (Fig. 12): the change in BC emissions is from -22.0% in May 2020 to 55.0% in September 2020, and the change in OC emissions is from -29.0% in May 2020 to September

2020 94.0%, while the change in dust emissions is from -57.1% in November 2019 to 52.9% in September 2020. Biomass burning, one of the main sources of BC and OC, can strongly vary daily, monthly and seasonally, as do dust emissions, and these variations are reflected in the BC, OC and dust variability seen in Figs. 8a, 8b and 8c. In addition, the interannual variations of BC, OC and dust emissions are also large. For example, the dust emissions in September 2019 are 38.2% lower than the average level, but the dust emissions in September 2020 are 52.9% higher than the average level (Fig. 12). This is to

say, it is basically impossible to roughly predict the change of aerosol mass in the next year by using the change trend of aerosol mass in the previous year.

The study of monthly and interannual variations in aerosol mass is important because it determines whether it is appropriate to use aerosol climatology fields rather than aerosol prognostic fields in weather forecasting to save computational resources. Unfortunately, the strong correlation between aerosol mass and emissions and the uncertainty of natural sources in aerosol

emissions lead to irregular aerosol changes in the atmosphere on a global scale. Therefore, when considering the effect of aerosol radiation on the weather system, such as in FV3GFS, the prognostic aerosol fields predicted by the chemical part of the meteorological model should be used. Otherwise the errors introduced by aerosols may outweigh the benefits of coupling chemical and meteorological models.

### 3.11. Annual Budget

The total annual aerosol emissions and annual average burdens (unit: Tg) were summarized in Table 2. The numbers in parentheses are from NASA GEOS4-GOCART [Colarco et al., 2010]. For both BC and OC, the total annual emissions in GEFS-Aerosols are larger than those in GEOS4-GOCART, especially the OC emissions, which are approximately 60% larger. GEFS-Aerosols uses GBBEPx as the biomass emissions inventory and GEOS4-GOCART uses the Global Fire Emissions Database Version 2 (GFEDv2). Because GBBEPx combines MODIS (Moderate Resolution Imaging Spectroradiometer) and

VIIRS (Visible Infrared Imaging Radiometer Suite) fire observations during this study period, it tends to have larger inventories than GFED. For the removal process, the two models have similar percentages from wet and dry deposition. Despite the larger emissions of BC and OC in GEFS-Aerosols, their burdens were smaller, suggesting that the BC and OC removal process was much faster in GEFS-Aerosols than in GEOS4-GOCART. This may be caused by a larger wet removal scaling factor in GEFS-Aerosols, since wet deposition accounts for about 70% of the total removal process for these2 types of aerosols,

Alternatively, the assumption that hydrophobic BCs and OCs also undergo wet deposition in GEFS-Aerosols may need to be reconsidered.

For dust, total emissions in GEOS4-GOCART (1970 Tg) are approximately four times higher than in GEFS-Aerosols (490 Tg). This is mainly due to the different dust schemes used in each model; GEOS4-GOCART follows Ginoux's dust sources spatial distribution and magnitude [Ginoux et al., 2001], and the particle size-dependent wind threshold velocity to initiate dust

emission is from Marticorena and Bergametti [1995]; GEFS-Aerosols uses the Fengsha dust scheme [Dong et al., 2016], in which the threshold values are based on surface and wind tunnel flux measurements of saltation [Gillette, 1988] and a new sediment supply map, the Baker-Schepanski Map, is used [Zhang et al., 2022b]. If gravitational settling is included with dry

deposition in GEFS-Aerosols, the relative amount of wet vs dry deposition of the two models are very similar. However, the larger dust load gap (6x) between GEOS4-GOCART and GEFS-Aerosols reiterates that the aerosol removal process are much

faster for GEFS-Aerosols.

For sea salt, the annual emissions from GEFS-Aerosols are very similar to GEOS4-GOCART because the sea salt emissions schemes used in both models are similar [Gong, 2003]. For sea salt removal, GEFS-Aerosols showed a ratio of wet to dry deposition of approximately 6:4, in contrast to GEOS4-GOCART (4:6). The sea salt burden in GEFS-Aerosols was 340% smaller than that in GEOS4-GOCART. For sulfate, the GEFS-Aerosols results are close to the GEOS4-GOCART results.

In general, the atmospheric mass of aerosols in GEFS-Aerosols is less than that in GEOS4-GOCART because the aerosol removal process are faster in GEFS-Aerosols. Therefore, the aerosol lifetimes in GEFS-Aerosols are shorter than those in GEOS4-GOCART.

The budget analysis in Table 2 again demonstrates that two models can have completely different sources and sinks but end up with very similar concentration predictions, while at the same time it is difficult to discover which model is more correct.

Because few observational are available for verification, especially for aerosol removal processes or net mass fluxes at surfaces. Measuring aerosol deposition fluxes is extremely challenging [Farmer et al., 2021], so such observations are rare. For example, the in-cloud mass scavenging efficiency of BC [Yang et al., 2019], the number of studies in this area is small but the reported data vary greatly, making it difficult to use for model evaluation.

## 4.   Conclusions

A GEFS-Aerosols simulation was conducted from September 1, 2019 to September 30, 2020 to evaluate the model performance of GEFS-Aerosols. The purpose of this study was to understand how aerosol chemical and physical processes affect ambient aerosol concentrations by placing aerosol wet deposition, dry deposition, reactions, gravitational deposition, and emissions into the aerosol mass balance equation (Equation. 1), and also to evaluate how realistic the model budget was by comparing the model budget to other model analyses.

First, the conservation of aerosol mass in GEFS-Aerosols simulations and model analyzes is examined. GEFS-Aerosols has a very small (< 0.13%) mass non-conservation when processing the initial input to the model, which is caused by the aerosol input not being updated correspondingly when the meteorological input is updated. At the same time, in the GEFS-Aerosols analysis, due to the limitation of the temporal resolution of the output of the GEFS-aerosol model in the analysis and the linearity assumption of aerosol physical and chemical processes, there are also subtle errors (< -0.033%) in the analysis results

of the model. Fortunately, these errors do not accumulate in the annual GEFS aerosol simulations and analyses, as positive errors cancel out negative ones.

In these particular GEFS-Aerosols simulations, sea salt contributes the most aerosol globally, followed by dust, OC, sulfate, and BC. The total mass of aerosols in GEFS-Aerosols is highly correlated with their emissions, with the exception of sulfate. Because there is no direct sulfate emission in GEFS-Aerosols, as sulfate is produced by the atmospheric oxidation of $SO_2$ and

445 DMS. Although biomass burning also emits $SO_2$, anthropogenic $SO_2$ dominates its emissions globally, accounting for more than 99%. Consequently, the interannual variations of sulfate mass in GEFS-Aerosols is minimal, and the sulfate mass trend is nearly stable over the year. This is in contrast to the decreasing trends in BC and OC and increasing trends in dust and sea salt observed over the simulated period. The strong monthly and interannual variations of natural sources of BC, OC and dust in GEFS-Aerosols indicates that prognostic fields need to be used as input when aerosol radiative feedback is to be applied to

450 weather forecasting and climate prediction.

 In GEFS-Aerosols, aerosol wet deposition is greater than aerosol dry deposition. The gravitational settling of dust and sea salt are also included in the model. For the coarse aerosol mode, gravitational settling is the main removal process, such as in DUST4, DUST5, SS4 and SS5. Aerosol removal is also highly correlated with aerosol emissions. For dust and sea salt, the aerosol size distribution in emissions determines the contribution of aerosols in each size to total aerosol removal as well as

455 the composition of aerosols in the atmosphere, suggesting that the size (bin) distribution of emissions is as important as their total volume.

 Aerosol particle size (bin) affects not only its contribution in aerosol removal, but also the efficiency of aerosol removal from the atmosphere, thereby determining the lifetime of aerosols in the atmosphere and ultimately determining the global aerosol distribution. From the mass and removal of the aerosol, the lifetime of the aerosol in the atmosphere was calculated. The

460 lifetimes of BC, OC, dust, sea salt and sulfate are 4.06, 4.29, 4.59, 0.34 and 3.3 days, respectively. The lifetime of the aerosol determines the vertical distribution of the aerosol. On average globally, for sea salt, due to its short lifetime, its mass accumulates on average around 800 hPa. For other species, more mass accumulates between 800 hPa and 600 hPa, even at heights above 400 hPa, significant mass of BC2, OC2 and sulfate can be simulated in the model.

 GEFS-Aerosols results were compared with GEOS4-GOCART in terms of total annual emissions, aerosol loading, and aerosol

465 lifetime. Excluding differences in model resolution and simulation time, GEFS-Aerosols simulated more BC and OC emissions, much lower dust emissions, and nearly similar sea salt emissions. The faster removal process in GEFS-Aerosols, most likely due to fast wet removal, resulted in a lower aerosol burden in GEFS-Aerosols than in GEOS4-GOCART. The lifetime of aerosols in GEFS-Aerosols is shorter than that in GEOS4-GOCART.

 Through this study, there are three findings that can be used to improve GEFS-Aerosols: 1) Correct the model's mass non-

470 conservation error when dealing with initial conditions; 2) Extend the residence time of aerosol in the atmosphere in the model. 3) Improve estimates of total aerosol emissions and particle size distribution in emissions.

## 5. Code and data availability

GEFS-Aerosols operational code used in this study is available at DOI: 10.5281/zenodo.7930284

## 6.    Author contributions

LP was one of the initiators and the actual implementer of this research. PB evaluated the simulation results in GEFS-Aerosols; LZ, RM, GG and SM were the developers of the chemical models in GEFS-Aerosols; BB provided the anthropogenic emissions used in GEFS-Aerosols and was responsible for the dust scheme in GEFS-Aerosols. JM was not only one of the initiators of this research, but also responsible for the implementation of GEFS-Aerosols from research to operation. SK and XZ supplied biomass burning emissions to GEFS-Aerosols. FY was the lead developer of the meteorological models (FV3GFS) used in GEFS-Aerosols. The aerosol model becoming a member of GEFS was the efforts of GF and IS.

## 7.    Competing interests

None of the authors have any competing interests

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

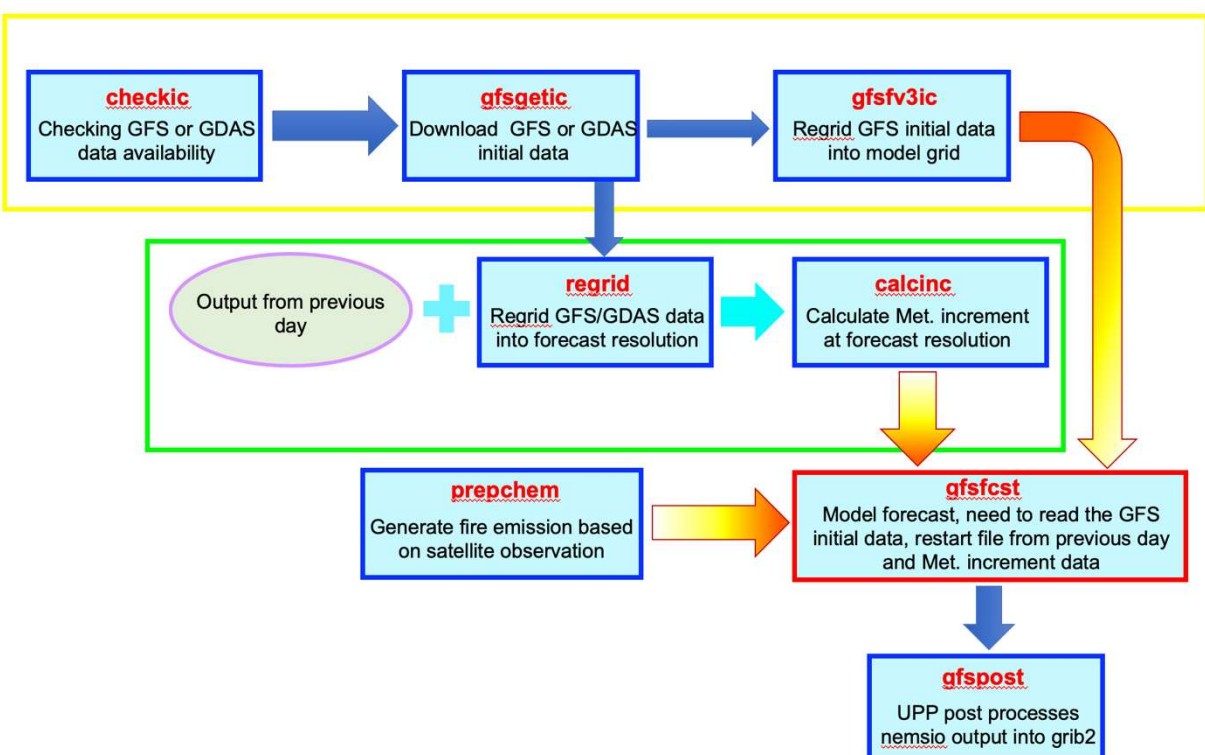

Figure 1: flowchart of GEFS-Aerosols forecast workflow (courtesy of Zhang et al., 2022b).

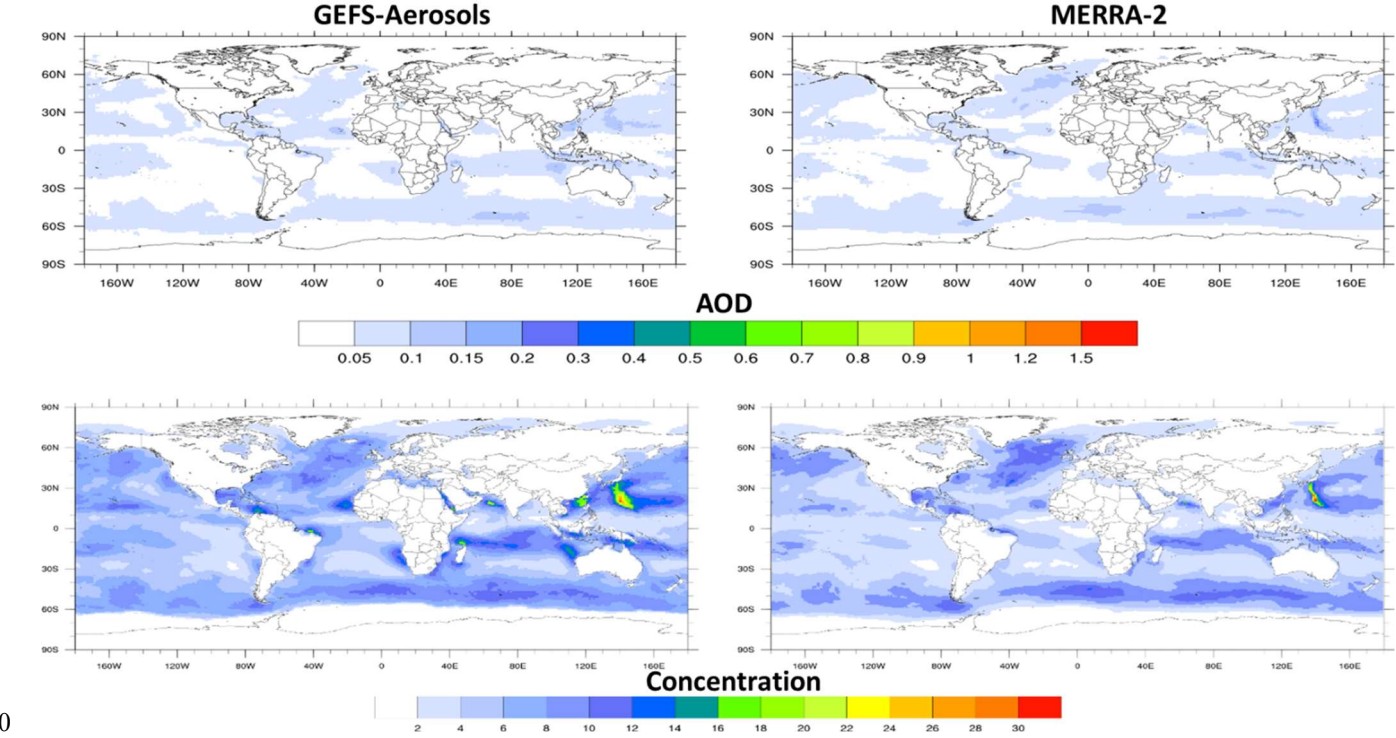


Figure 2: the monthly mean sea salt AOD (top) and surface mass concentration (μg/m³) (bottom) simulated in October 2019 from GEFS-Aerosols (left) and MERRA-2 (right).



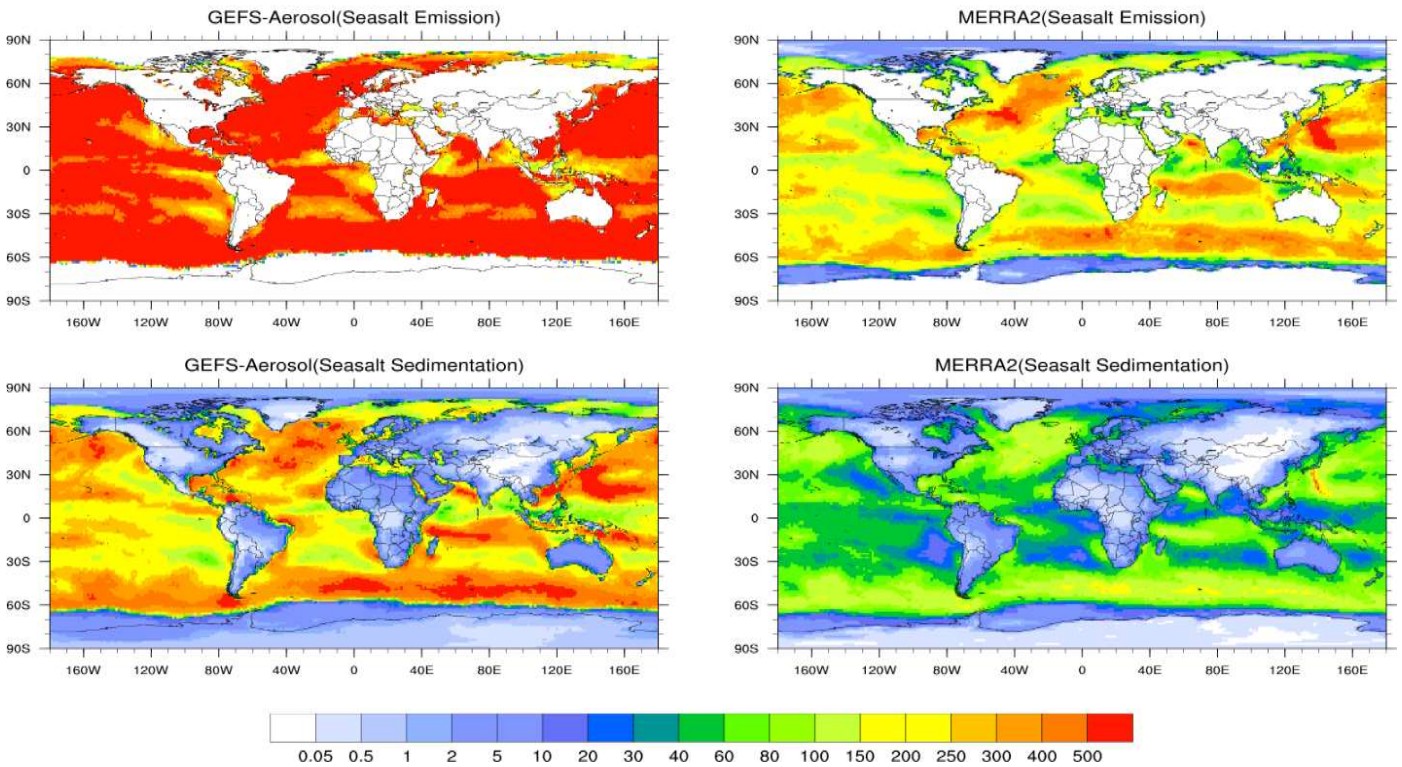

Figure 3: sea salt emission and sedimentation (October, 2019) in ng/m²/s for GEFS-Aerosols (left side) and MERRA-2 (right side).




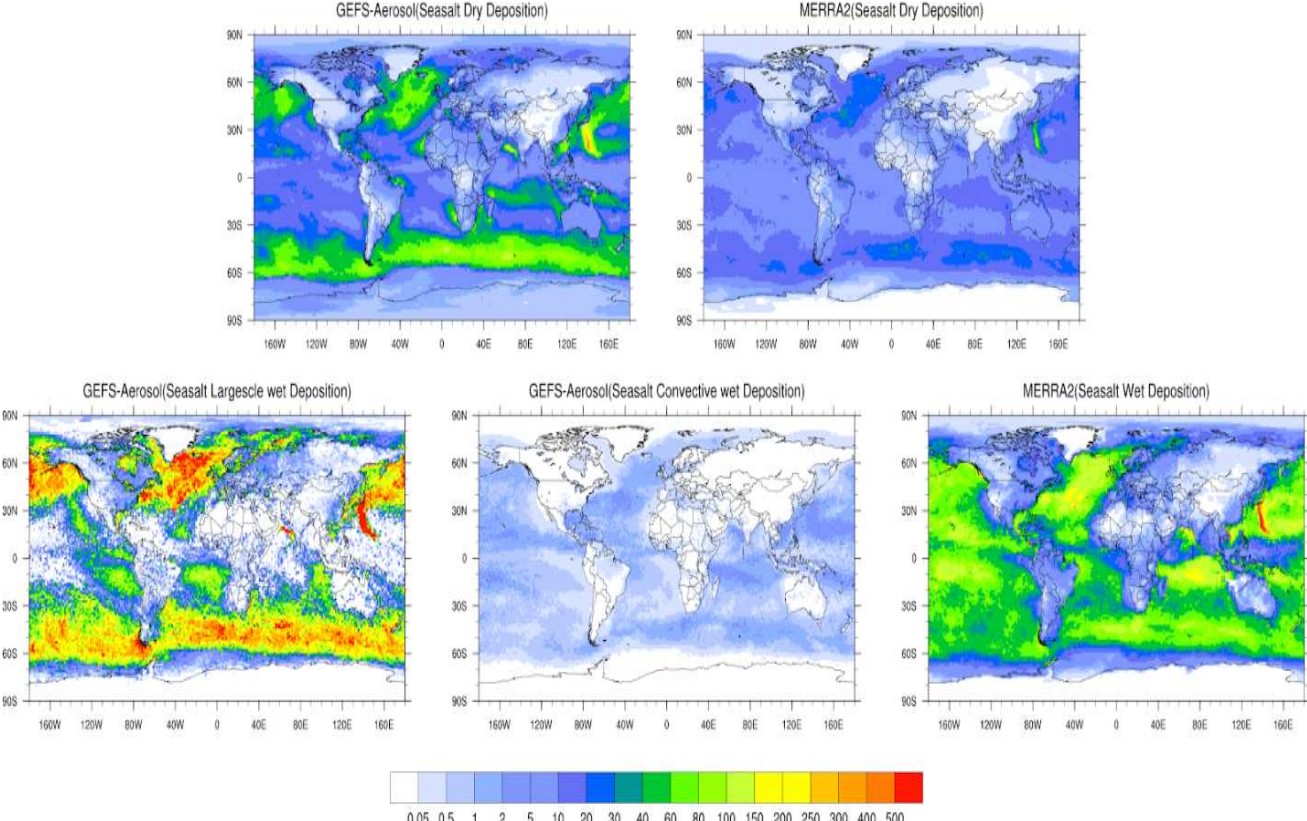

Figure 4: sea salt Deposition (October, 2019) in ng/m$^2$/s for GEFS-Aerosols (left side) and MERRA-2 (right side).


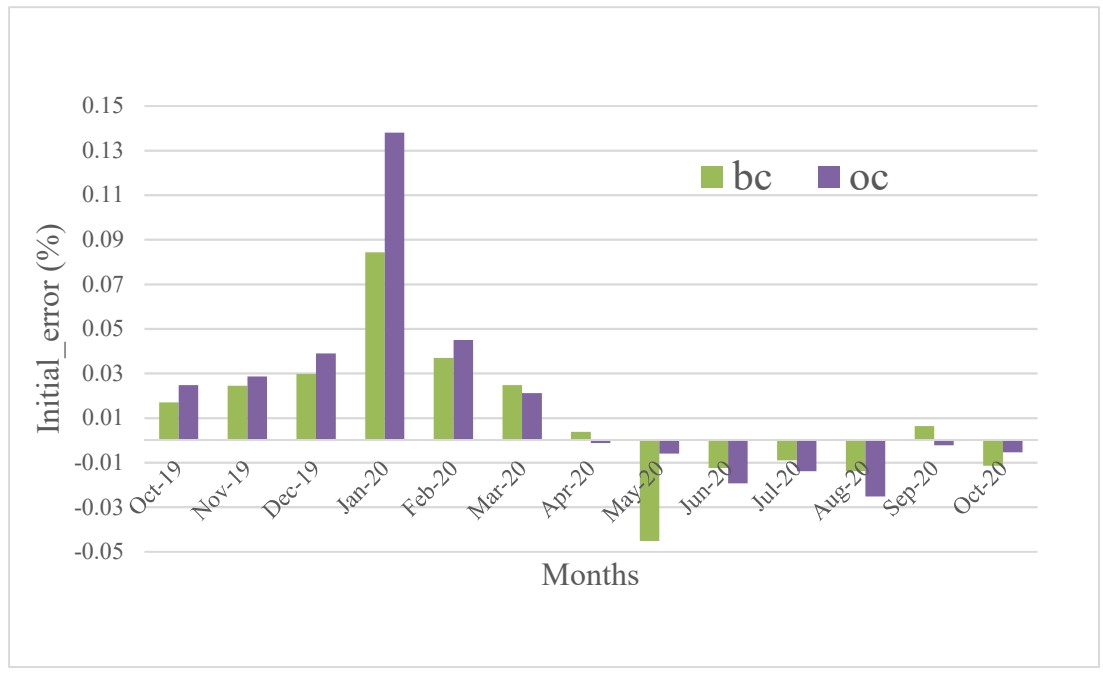


Figure 5: BC and OC initial errors (%) at the beginning of each month for the GEFS-Aerosols simulation time period from October 2019 to October 2020.

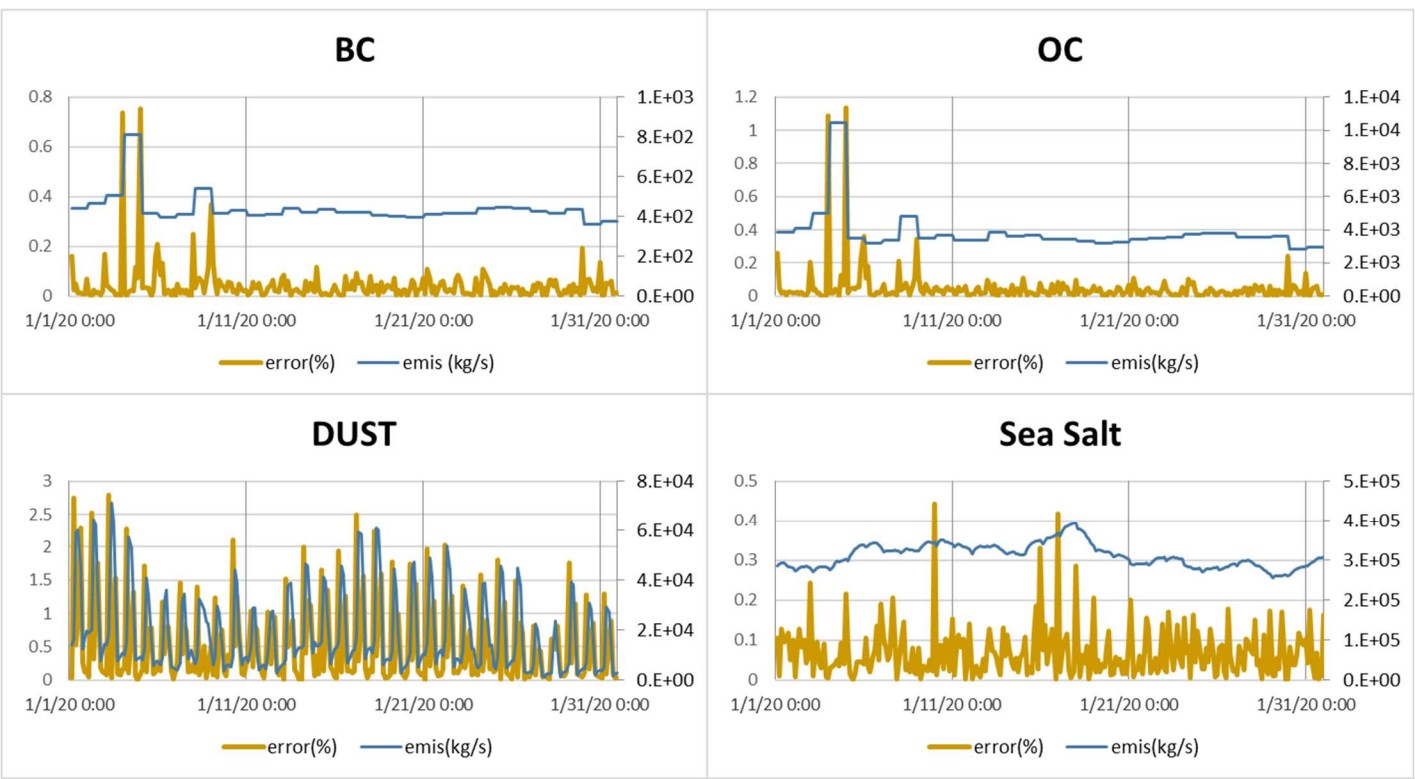

Figure 6: absolute value of model error (brown line, Primary Axis) for BC, OC, DUST and Sea Salt during GEFS-Aerosol simulations in January 2020, and corresponding aerosol emissions (blue line, Secondary Axis).



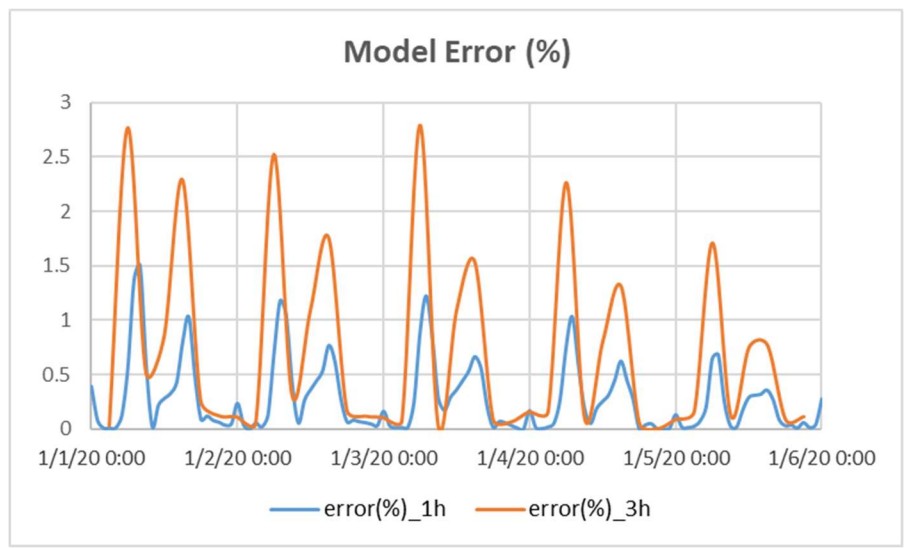

Figure 7: absolute value of DUST model error (%) at the different GEFS-Aerosols output frequencies for the 5-day forecast
from January 1, to January 5, 2020: every 3 hours (orange) and every 1 hour (blue).


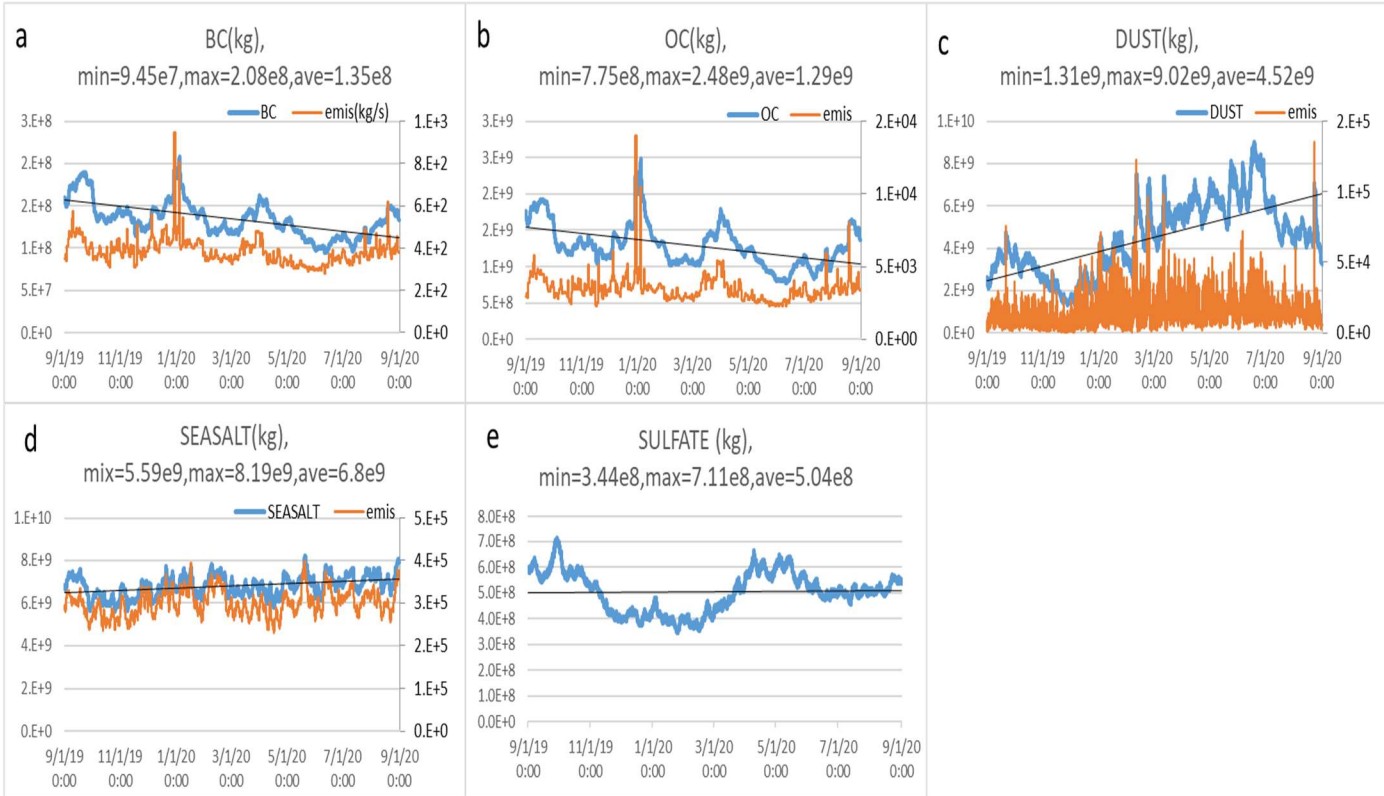

Figure 8: aerosol mass changes of BC (a), OC (b), dust (c), sea salt (d) and sulfate (e) in the atmosphere from September 1, 2019 to September 1, 2020. The blue line (primary axis) represents aerosol mass (kg), the orange line (secondary axis) represents aerosol emission (kg/s), and the black line represents the trend of aerosol mass.


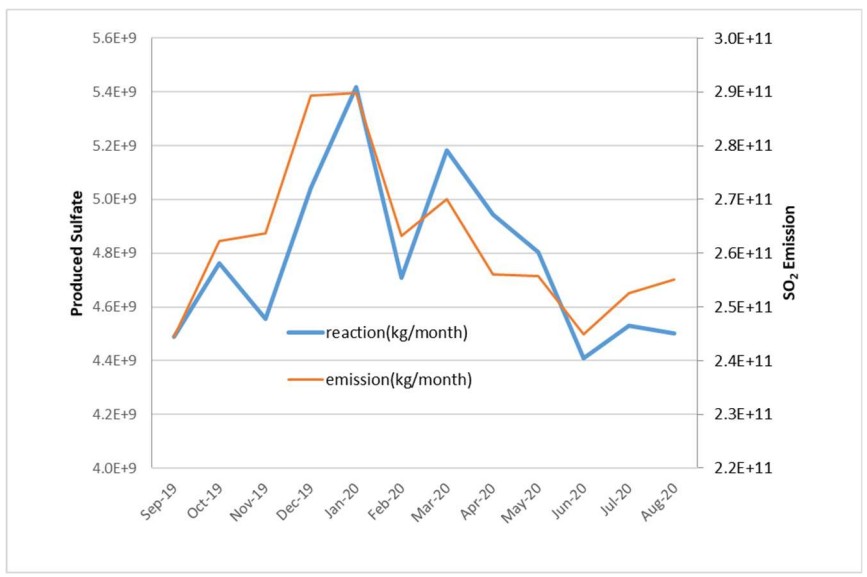


Figure 9: monthly sulfate from reactions (orange line, primary axis) and SO₂ emissions (blue line, secondary axis) in the GEFS-Aerosols simulation (kg/month).



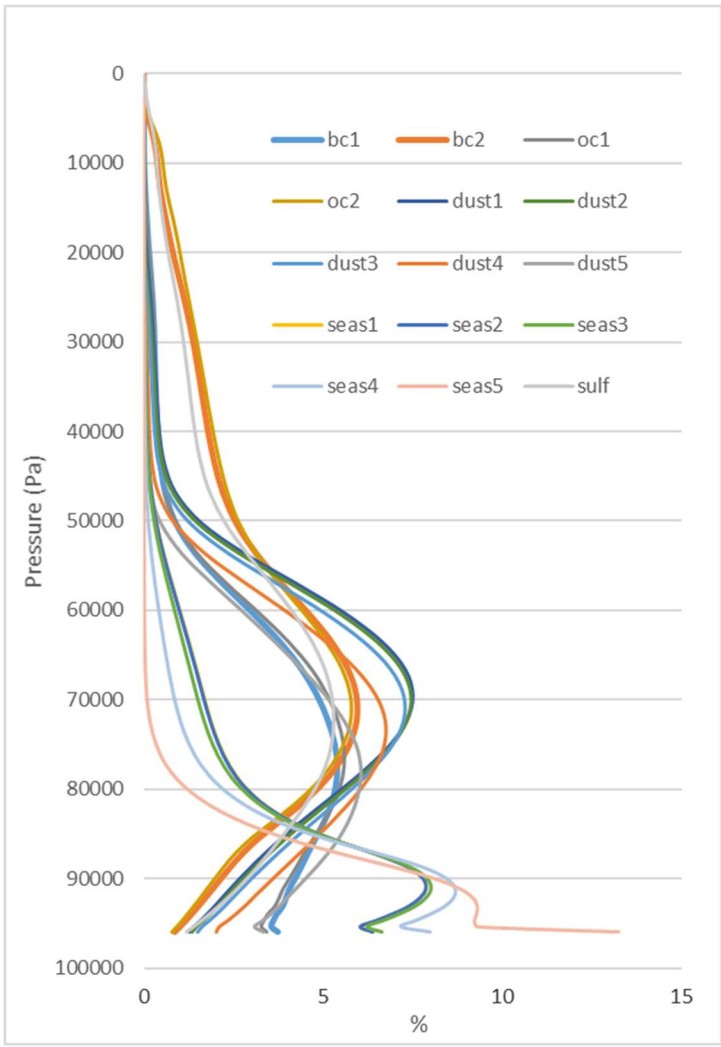

Figure 10: vertical distribution of aerosol mass percent along the pressure for the 64 model layers in GEFS-Aerosols, the data used in this figure are from September 2019 monthly averages and are global averages.

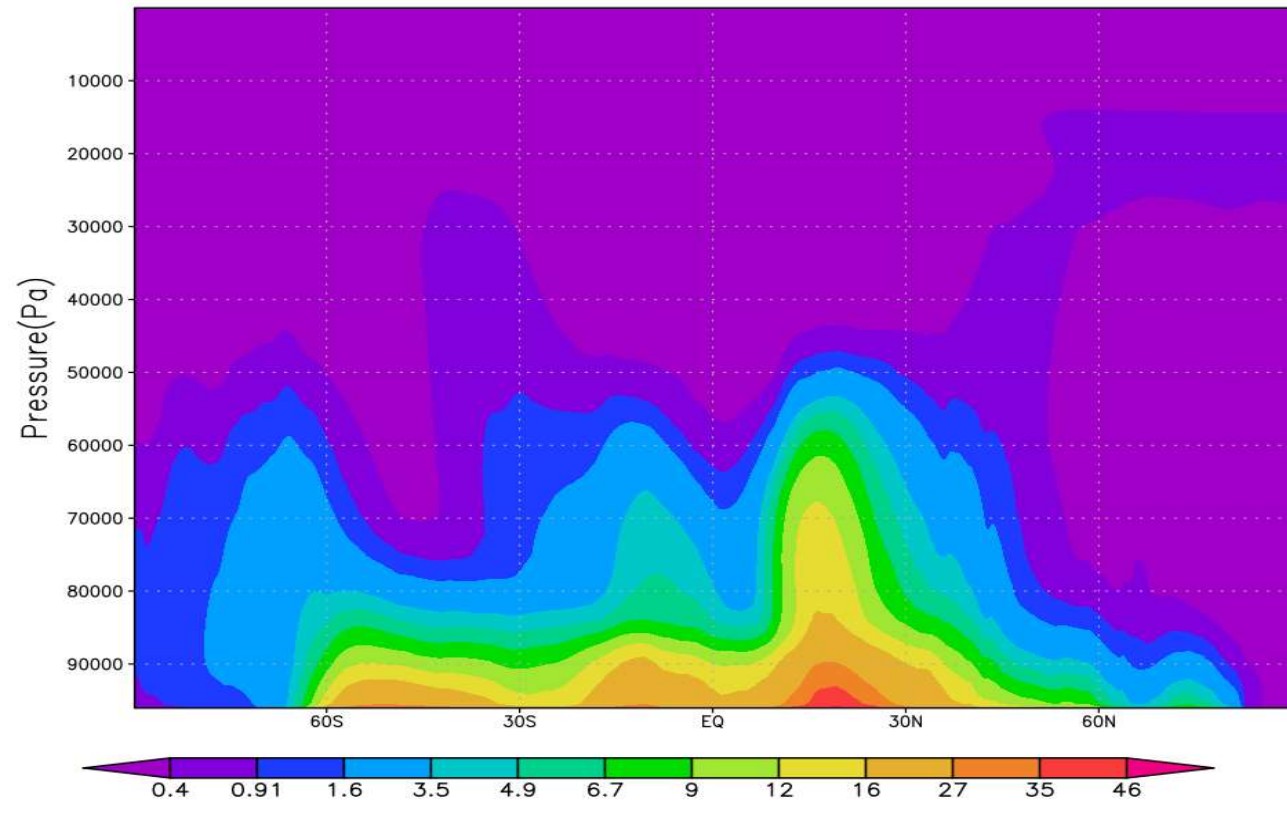

Figure 11: zonal and monthly mean (September, 2019) simulated total aerosol mix ratio (µg/kg_air).

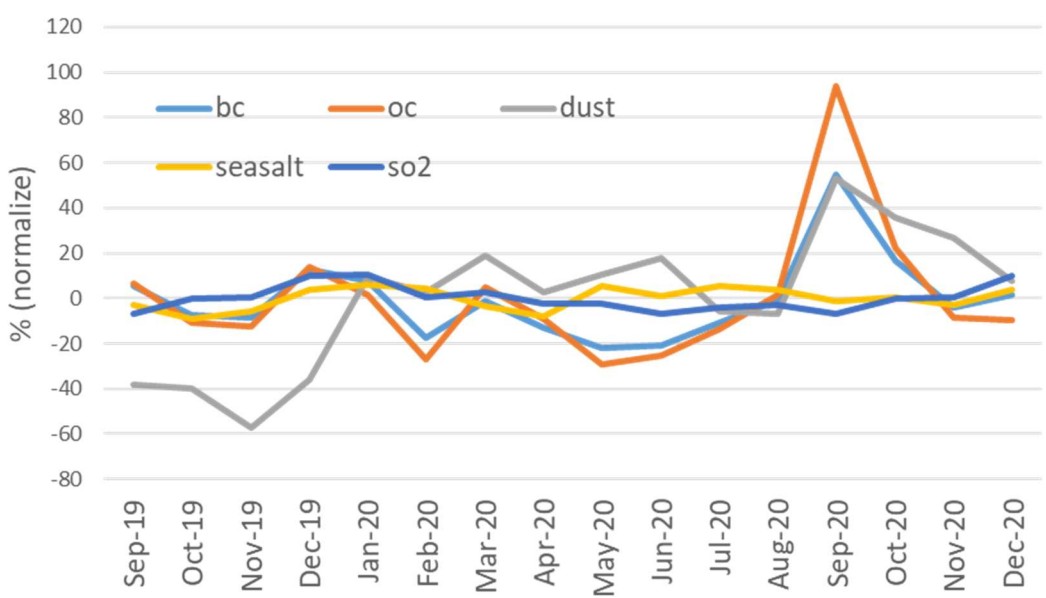

Figure 12: normalized monthly variations (%) in GEFS-Aerosols emissions of BC, OC, dust, sea salt and $SO_2$ from September 2019 to December 2020



| Aerosols | Bins (micron) | Emission (%) (anthropogenic) | Sediment (%) | Wet deposition (%) | Dry deposition (%) | Total remove (%) | Composition (%) | Lifetime (day) |
|---|---|---|---|---|---|---|---|---|
| **BC1** | Hydrophobic | 100 (66.9 [30.2-84.5]) | n/a | 43.04 | 72.14 | 50.53 | 30.33 | 4.06 |
| **BC2** | Hydrophilic | 0 | n/a | 56.96 | 27.86 | 49.47 | 69.67 | |
| **OC1** | Hydrophobic | 100 (50.0 [13.4-73.5] ) | n/a | 40.59 | 72.05 | 50.53 | 30.33 | 4.29 |
| **OC2** | Hydrophilic | 0 | n/a | 59.41 | 27.95 | 49.47 | 69.67 | |
| **DUST1** | 0.1-1.0 | 10.20 | 0.70 | 20.86 | 14.93 | 10.28 | 20.29 | 6.89 |
| **DUST2** | 1.0-1.8 | 10.19 | 2.45 | 18.72 | 14.22 | 10.26 | 18.14 | 6.16 |
| **DUST3** | 1.8-3.0 | 20.90 | 12.60 | 29.48 | 25.90 | 21.01 | 28.78 | 4.77 |
| **DUST4** | 3.0-6.0 | 48.46 | 65.02 | 29.53 | 40.87 | 48.33 | 31.06 | 2.26 |
| **DUST5** | 6.0-10.0 | 10.26 | 19.22 | 1.41 | 4.09 | 10.12 | 1.73 | 0.61 |
| **Seasalt1** | 0.03-0.1 | 0.03 | 0.00 | 0.05 | 0.05 | 0.03 | 0.07 | 0.53 |
| **Seasalt2** | 0.1-0.5 | 1.10 | 0.01 | 1.73 | 1.70 | 1.10 | 2.17 | 0.52 |
| **Seasalt3** | 0.5-1.5 | 11.70 | 1.32 | 17.75 | 17.43 | 11.70 | 21.33 | 0.48 |
| **Seasalt4** | 1.5-5.0 | 49.79 | 28.57 | 62.19 | 61.09 | 49.79 | 63.71 | 0.34 |
| **Seasalt5** | 5.0-10.0 | 37.38 | 70.10 | 18.28 | 19.73 | 37.38 | 12.73 | 0.09 |

Table 1: Summary of emission, sediment, deposition, composition and lifetime of BC, OC, dust and sea salt in GEFS-Aerosols.


| Aerosols | Wet (%) | Dry (%) | Sediment (%) | Wet (Tg/Y) | Dry (Tg/Y) | Sediment (Tg/Y) | Emission (Tg/Y) | Burden (Tg) | Lifetime (day) |
|---|---|---|---|---|---|---|---|---|---|
| BC | 75 (68.4) | 25 (31.6) | N/A | 9.1 | 3.1 | N/A | 12.2 (10.06) | 0.14 (0.24) | 4.06 (8.82) |
| OC | 75 (71.7) | 25 (28.3) | N/A | 82.7 | 27.4 | N/A | 111 (68.76) | 1.29 (1.30) | 4.29 (6.90) |
| DUST | 36.5 (32.2) | 17.3 (67.8) | 46.3 (N/A) | 182 | 84.2 | 223 | 490 (1970) | 4.52 (31.6) | 4.59 (5.85) |
| Sea Salt | 59.0 (40.0) | 4.3 (60.0) | 36.7 (N/A) | 5590 | 40.6 | 3470 | 9470 (9729) | 6.80 (23.4) | 0.34 (0.88) |
| Sulfate | 90.5 (85.5) | 9.5 (14.5) | N/A | 0.00068 | 0.0065 | N/A | 0 | 0.51 (0.71) | 3.3 (4.42) |

Table 2: summary of total annual aerosol emissions and annual average burdens (Tg), the numbers in parentheses are from NASA GEOS4-GOCART [Colarco et al., 2010].
