# Peer review of "Analysis of GEFS-Aerosols annual budget to better understand the aerosol predictions simulated in the model"

_Geoscientific Model Development, 2023_

## Author Comment (AC1)

The authors would like to thank the reviewers for volunteering their time to review this manuscript. Your comments make this manuscript better and better. I have carefully read your valuable suggestions, and the following is my reply.

*This manuscript described a process-based budget analysis of the GEFS-Aerosols*
*chemical transport model, including the processes of emissions, reactions and removal. This model budget analysis includes the comparison to the MERRA-2 and GEO4-GOCART, but has few verification with observations, making it hard to evaluate which process has big uncertainties.*

Bhattacharjee et al., 2023 (DOI: https://doi.org/10.1175/WAF-D-22-0083.1) evaluated the
simulation results of the GEFS-Aerosols model using AOD data derived from satellite retrieval (MODIS and VIIRS), AOD data simulated by other models (MEERA2 and NGAC), and AOD data observed from 50 AERONET stations. The period of evaluation from August 2019 to August 2020 almost coincides with the time period of this study, namely from September 2019 to September 2020. In addition to the regular daily or monthly forecast
evaluations of GEFS-Aerosols, three special events were also utilized to evaluate the performance of GEFS-Aerosols. These include dust events in Northwest Africa, agricultural fires in northern India and the August fire complex in northern California.

Zhang et al., 2022 (DOI: https://doi.org/10.5194/gmd-15-5337-2022) evaluated not only the AOD simulated by GEFS-Aerosols from 5 July to 30 November 2019, but also the aerosol
concentrations simulated by GEFS-Aerosols during the 22-month ATOM (Atmospheric TOmography Mission) period from 2016 to 2019.

Those sentences has been added to lines 40 to 48 of the manuscript.

The following is the author's response to the detailed comment

*Section 2.1 and 2.2. The mass balance equation and associated processes did not*
*mention the model's advection, diffusion, and physical processes. How well has the aerosol mass been conserved in these processes? What's this model's top boundary treatment, and how does the model control the mass leakage through the domain top?*

The research object of mass balance equation is the total mass of aerosol in the
atmosphere. Any aerosol mass fluxes (e.g. emissions and removals) leaving or entering the system are considered in the mass balance equation. The reaction is also included in the equation as it changes the aerosol species and the mass balance equation is calculated for the aerosol species. Although aerosols are affected by advection, diffusion, and physical processes, these processes are not specifically considered in the equations because they
do not cause aerosols to leave/enter the system or change aerosol species. However, these processes do change the concentration of aerosols in the atmosphere, and this effect is ultimately reflected in the two terms "initial and final" in the mass balance equation.

The pressure at the top of the model in GEFS-Aerosols is 200 Pa. As shown in Figures 10 and 11, it is almost impossible for aerosols to be elevated to this level in our simulations, and therefore, at this level, the aerosol concentration in GEFS-Aerosols is the background concentration (1 x 10$^{-16}$ µg/kg).

*The gravitational settling of aerosols is usually applied to mass movement from upper layers to the lowest layer of the model, which won't affect the total mass. The*
*removal of aerosol mass from the model's lowest layer to ground surface refers to dry deposition, which should include the gravitational sediment. Does the GEFS-Aerosol's dry deposition scheme exclude the gravitational sedimentation? Please clarify.*

The gravity scheme used in GEFS-Aerosol is mentioned in the manuscript lines 76 to 78 "
the computation of gravitation settling for dust and sea salt are based on the updated finite-difference scheme in WRF-CHEM (Ukhov et al, 2021)". This scheme not only calculates the gravity settlement from the upper layer of the model to the lower layer of the model, but also calculates the gravity settlement from the bottom layer of the model to the ground. Meanwhile, dry deposition in GEFS-aerosol was calculated by dry deposition velocity based
on aerodynamic resistance, sublayer resistance and surface resistance. Therefore, aerosol gravity deposition and aerosol dry deposition are completely separated in GEFS-Aerosols.

*Fig 2, and line 115, Are the sea salt AOD calculation method same in GEFS-Aerosol and MERRA-2? Besides AOD comparison, it is better to have mass concentration*
*comparison with observations for sea salt near sea surface. How about the mass flux for other species?*

In GEFS-Aerosols, AOD is calculated using a look-up table (LUT) of aerosol optical properties from the NASA GOCART model, so AOD calculations are consistent in GEFS-Aerosols and MERRA2.

[Figure]

Seasalt Surface mass conc. in mcg/m3 for Oct,2019

The figure above shows a comparison of surface concentrations of sea salt aerosols in GEFS-Aerosols and MERRA2 in October 2019.

*Fig 3, 4 and line 117-120. Similar question, does the dry deposition of the lowest*
*model layer exclude the sedimentation? It would better to show the net surface flux*
*of sea salt in GEFS-Aerosol and MERRA-2, to show whether their emission and*
*removal processes are balanced.*
It can be seen from Table 2 that the total amount of sea salt emissions and removal
processes in GEFS-Aerosols are basically balanced during the one year simulation period.

*Fig 6 and corresponding discussion around line 138-145. The model error for dust*
*mass is highly correlated with the dust emission, but these correlations do not exist*
*for BC, OC and sea salt. Any further discussion about the difference among these*
*species, which processes result in the difference?*
Indeed, the model errors for BC and OC are also highly correlated with emissions, as
indicated in lines 151–152 of the updated manuscript. "The larger model errors for dust, OC
and BC coincide with emission outbreaks for dust, OC and BC, which are shown as the
emission changes (kg/s -> blue line) in Fig. 6".
However, the model errors for dust are larger than those for BC, OC and sea salt. The
reasons for this were discussed in lines 159 to 164 of the updated manuscript "The
assumptions of linearity in aerosol deposition and emissions in the GEFS-Aerosols
calculations are the main cause of its model error. This assumption only affects deposition
calculations for BC and OC (The daily emissions of BC and OC are constant.), but for dust
and sea salt it affects not only deposition but also emissions calculations. At the same time,
the wind threshold velocity makes the dust emissions more nonlinear than the source or
sink terms of the other aerosol types. Therefore, the model errors for dust and sea salt are
higher than those for BC and OC, while the model errors for dust are the highest."

*Fig 10. What is the temporal and spatial extents of these profiles? Please clarify. It is*
*better to separate the profiles over ocean and land, and have a deeper discussion for*
*sea salt and land-source aerosols etc.*

*Line 282-290. It is more convincing if any direct comparison with observations can be*
*included here, like CALIOP or in-situ measurements.*

The data used in Fig. 10 are from September 2019 monthly averages and are global
averages. Similar graphs were plotted for other months in this experiment. Since these
figures are very similar, they are not shown in this manuscript.

Both Fig. 10 and Fig. 11 represent the simulated vertical distribution of aerosols in GEFS-
Aerosols. By comparing Fig. 10 and Fig. 11, we noticed that the different data processing
methods when plotting Fig. 10 and Fig. 11 made the GEFS-aerosol vertical profiles in Fig.
10 and Fig. 11 very different. Fig. 11 shows that aerosol concentrations decrease with
increasing altitude. However, Fig. 10 shows that at least for some aerosols (e.g. BC, OC,
sulfate, and dust) concentrations are higher at higher altitudes. Even so, they are not
contradictory. Because Fig. 11 (zonal distribution) is more representative of the vertical
distribution of aerosols near the source area (e.g. land), while Fig. 10 is more representative
of the vertical distribution of aerosols away from the source area (e.g. over the ocean).
Therefore, to evaluate Fig. 11, the HIPPO experiment and CALIOP observations were used,
since they measure vertical profiles of aerosols in remote regions.

Added the following sentence into the manuscript: "The decrease in aerosol concentration with increasing altitude shown in Fig. 11 is significantly different from that shown in Fig. 10. Fig. 10 shows that certain aerosols (e.g. BC, OC, sulfate and dust) are more concentrated at higher altitudes. However, Fig. 10 and Fig. 11 are not contradictory. Because Fig. 11 is more representative of the vertical distribution of aerosols near the source (such as over land), while Fig. 10 is more representative of the vertical distribution of aerosols far away from the source (such as over the ocean). Therefore, in the validation of Fig. 10, the HIPPO experiment and CALIOP observations were used because they measured the vertical profile of aerosols in remote areas."

***Table 2 and Section 3.11 It is better to make consistent by changing the units of wet/dry deposition and sediment from percent to Tg/Year, comparable to emission etc.***

Three more columns were added to Table 2, indicating the amount of aerosol removed by wet deposition, dry deposition, and deposition (Tg/Y) in GEFS-aerosol, and sentence "and the total removal of aerosols (i.e. the sum of wet, dry and sedimentation) is almost equal to their total emissions (Table. 2)." was added to the text.

***Line 350-365. Giving the big discrepancies between the models, is there any observation available to verify aerosol removal process or surface net mass flux?***

Measuring aerosol deposition fluxes is extremely challenging (Farmer et al., 2021), so observations are limited. For example, the in-cloud mass scavenging efficiency of BC (Yang et al., 2019), the number of studies in this area is small but the data reported vary widely, making them difficult to use for model evaluation.

In GEFS-Aerosols: "The GOCART dry deposition protocol [Chin et al., 2000] was used for GEFS-Aerosols. Wet deposition in GEFS-Aerosols is the sum of large-scale wet removals and convective scavenging. The large-scale wet removal scheme is from WRF-CHEM (https://github.com/wrf-model/WRF/blob/master/chem/module_wetdep_ls.F), and the convective scavenge is calculated in FV3GFS physics and is based on the simplified Arakawa-Schubert (SAS) scheme [Pan et al, 1994; Zhang et al., 2022b]. The computation of gravitational settling for dust and sea salt are based on the updated finite-difference scheme in WRF-CHEM [Ukhov et al., 2021]."

*Farmer, D. K., Boedicker, E. K., & DeBolt, H. M. (2021). Dry deposition of atmospheric aerosols: Approaches, observations, and mechanisms. Annual review of physical chemistry, 72, 375-397.*
*Yang, Y., Fu, Y., Lin, Q., Jiang, F., Lian, X., Li, L., ... & Sheng, G. (2019). Recent advances in quantifying wet scavenging efficiency of black carbon aerosol. Atmosphere, 10(4), 175.*

---

## Author Comment (AC2)

*My major comments:*

The authors would like to thank the reviewers for volunteering their time to review this manuscript. Your comments make this manuscript better and better. I have carefully read your valuable suggestions, and the following is my reply.

*-The authors plainly describe what they see, not what they learn from the analysis.   The manuscript could be further improved if the authors present the results with more scientific insight.*

*-It is very difficult to verify simulated lifetime and annual emission and removal due to lack of observational evidence. The authors only compare the GEFS-Aerosols*
*results with GEOS4-GOCART from Colarco et al. 2010.  How about the AeroCom consensus?*

GEFS-Aerosols, a new global aerosol model developed by NOAA, became operational in September 2020, and the last study time of AeroCom Phase III was 2010, so GEFS-Aerosols did not have the opportunity to participate in AeroCom for inter-model comparison.
However, GEFS-Aerosols has been participating in GAFIS (global air quality forecasting and information system) ([https://community.wmo.int/en/activity-areas/gaw/science-for-services/gafis](https://community.wmo.int/en/activity-areas/gaw/science-for-services/gafis)) since 2022, a WMO (World Meteorological Organization) organized project for models inter-comparison.

*Here are some minor comments for the authors to consider:*

*-Line 36 'As a first step towards this goal, 'GEFS-Aerosols was implemented to replace NGAC.  The efforts to enable prognostic aerosol capability toward the goal started with the implementation of NGAC.  It is not clear to me why the authors view the GEFS-Aerosols implementation as the "first step."*

NGAC is an offline model that was replaced in September 2020 by the online model GEFS-
Aerosols. NCEP is continuing to develop the model, such as adding aerosol data assimilation to the system. The goal is to incorporate aerosol components into NOAA UFS (Unified Forecast System). As such, we see this replacement as the first step toward our ultimate goal.

*-Line 43-44: 'because these processes occur before the model output and they are*
*the determinants of aerosol concentration."   I agree that budget analysis is important to examine model's fidelity/performance.  However, the justification "these processes occur before the model output and they are the determinants of aerosol concentration" is very odd and weak.  Budget analysis can reveal whether the model have the bulk emission and removal processes right.  Whether these tendency*
*diagnostics are model output is totally irrelevant.*

I agree with the reviewer's comments. To clarify what I meant, the sentence has been changed to "because these processes are determinants of aerosol concentrations".

*-Line 40 'instead of focusing on aerosol concentration and aerosol optical depth (AOD) in a general aerosol evaluation'. Comparing GEFS-Aerosols model output with*

*PM/AOD observations is needed to thoroughly assess the model performance and identify potential model deficit. It is certainly all right for the authors to focus on budget analysis in this manuscript. Since the model vs observation evaluation has been conducted and reported in other papers [Lines 109-111], the authors should briefly describe the efforts.*

"Bhattacharjee et al. [2023] evaluated the simulation results of the GEFS-Aerosols model using AOD data derived from satellite retrieval (MODIS and VIIRS), AOD data simulated by other models (MEERA2 and NGAC), and AOD data observed from 50 AERONET stations. The period of evaluation from August 2019 to August 2020 almost coincides with the time period of this study, namely from September 2019 to September 2020. In addition to the regular daily or monthly forecast evaluations of GEFS-Aerosols, three special events were also utilized to evaluate the performance of GEFS-Aerosols. These include dust events in Northwest Africa, agricultural fires in northern India and the August fire complex in northern California. Zhang et al., [2022b] evaluated not only the AOD simulated by GEFS-Aerosols from 5 July to 30 November 2019, but also the aerosol concentrations simulated by GEFS-

Aerosols during the 22-month ATOM (Atmospheric TOmography Mission) period from 2016 to 2019." has been added to the text from lines 40 to 48.

*-Line53 Eq1: Initial + Emissions + Reactions = Final + Removal*

*Based on the governing equation, I'll probably present the equation as*

*Final = Initial + Emissions + Reactions – Removal*

The governing equation usually describe how a variable (such as aerosol concentration) changes when other variables change. Since this study focuses on the aerosol mass balance in GEFS-aerosols, the governing equation was converted to mass balance equation.

*-Line 78: 2.3 GEFS-Aerosols. Consider presenting this sub-section first in Section 2.*

It has been changed

*-Line 83: 'GOCART' Please define the acronym*

Added at line 24

*-Line 104: 'Fire Radiant Power (FRP) '. Fire Radiative Power?*

Corrected

*-Line 113: 'These processes ultimately define the aerosol concentration and AOD output by the model.' These processes ultimately determined 3-d aerosol*

*distribution, which in term affect concentration and AOD. But this sentence is somehow odd.*

It has been changed

*-Line 115 'MERRA2'  MERRA-2 is also based on GOCART. Does sea salt emission and removal scheme in GEFS-Aerosols differ from those in MERRA2?*

GEFS-Aerosols has the same sea salt emission mechanism as GEOS4-GOCART [Gong, 2003], but MERRA2 is based on GEOS5, which updated sea salt emission scheme [Randles et al., 2017]; the sea salt removal scheme in GEFS-Aerosols is also different from that in MERRA2.

Gong, S. L.: A parameterization of sea-salt aerosol source function for sub-and super-micron particles, Global biogeochemical cycles, 17(4), Doi10.1029/2003GB002079, 2003.

Randles, C. A., A. M., da Silva, V., Buchard, P. R., Colarco, A., Darmenov, R., Govindaraju, A., Smirnov, B., Holben, R., Ferrare, J., Hair, Y., Shinozuka and Flynn, C. J.: The MERRA-2 Aerosol Reanalysis, 1980 Onward. Part 1: System Description and Data Assimilation Evaluation, Journal of Climate, 30(17), 6823-6850, 10.1175/jcli-d-16-0609.1, 2017

*-Line 138 'Fig 6'   The principal behind the budget analysis is that aerosols net production is approximately equal to net loss when averaged over a long time (say multiple years).   It is not clear whether the monthly residual (Left side of Eq1 – Right side of Eq 1) should be interpreted as 'model error'.*

Annual aerosol deposition and sedimentation of BC, OC, dust and sea salt are added to Table 2, and as the reviewer states, aerosol emissions are almost equal to their total removals. We assume that the difference between the left side of Equation 1 and the right side of Equation 1 should be zero. If not, it means there is an error inside the model or in our analysis.

*-Line 155 "Therefore, the model errors for dust and sea salt are higher than those for BC and OC, while the model errors for dust are the highest.".    The text seems indicate that the model errors for dust and sea salt are caused by non-linearity in the emission/removal scheme.  This is not necessarily true.*

I should say yes, it might be true. If we could calculate dust and sea salt emissions and removals more precisely in our analysis, the model error could be very close to zero and much smaller than the numbers we saw in Fig 6 and Fig 7.

*-Line 159 'Global Aerosol Mass'.  It is insightful to specify when specific aerosol species reach max and min.  For instance, dust loading peaks in June and reached min in Nov.  This results are consistent with Africa dust activities.  However, it seems unnecessary for the authors to specify the exact date.*

I agree and corrected. The total amount of dust is highly correlated with the intensity of dust activity in Africa.

**-Line 172 'Annual trend'. How annual trend can be inferred from one-year simulation? Please clarify it.**

Corrected to "In the simulated year, the trends for BC and OC masses are decreasing (16.4% and 22.3%, respectively) and the trends for dust and sea salt are increasing (24.9% and 16.0%, respectively); for sulfates the trend is almost constant with only a very slight
decrease (8.09%)."

**-Line 181-186. The discussions about the partition can be presented in a table.**

The partition of dust and sea salt emissions is shown in Table 1.

**-Line 189 'Aerosol emissions are directly and indirectly related to their mass in the atmosphere'. Aerosol loading is certainly related to their emissions, and aerosol**
**emissions certainly affect aerosol mass. However, the statement is very awkward.**

The "indirect" mentioned in this sentence refers to sulfate, because there is no sulfate emission in GEFS-Aerosols. Sulfate is converted mainly from $SO_2$, which mainly comes from anthropogenic sources in GEFS-Aerosols.

**-Line 224-225: 'the size distribution of aerosol emissions becomes too important for**
**the removal process in GEFS-Aerosols simulations when the aerosol particle size is not changed in the model' Please clarify this sentence.**

For example, if the total dust emission is 50kg, of which dust1 emission is 5 kg, dust2 emission is 5 kg, dust3 emission is 10 kg, dust4 emission is 25 kg, dust4 emission is 5 kg, then finally 5 kg of dust is removed as dust1, 5 kg of dust is removed as dust2, 10 kg of dust
is removed as dust3, 25 kg of dust is removed as dust4, and 5 kg of dust is removed as dust5. In summary, for each dust size, the amount emitted is the amount removed since the particle size of the dust does not change in the GEFS-Aerosols simulation.

On the other hand, 100% OC is emitted as hydrophobic, but during the removal process, 50.5% OC is removed as hydrophobic and 49.5% OC is removed as hydrophilic, because
hydrophobic OC can be converted into hydrophilic OC.

**-Line 229 'as they do not undergo a size (bin) change.' The GOCART is a bulk mass scheme. It's not clear to me why the authors expect bin change.**

Please refer to the previous reply.

**-Line 265 'GOCART' Presume it's GOES4-GOCART. It does not hurt to make it clear.**

Corrected

*-Line 263-269, The differences in lifetime between GEFS-Aerosols and GEO4-GOCART are attributed to model resolution and simulation period. Both model use GOCART scheme. The differences and similarities between the two GOCART schemes should be considered. The difference between the two host AGCMs should also be discussed. If identical emission and removal scheme are implemented in both GEFS and GEOS4, the emissions and removal fluxes from the two model will still be different. The model with more active moisture process may produce more wet removal. The model with more noisy wind field may produce higher dust emissions.*

The similarities and differences between GEFS-Aerosol and GEOS4-GOCART in terms of aerosol deposition and emissions have been discussed in Section 2.1 "GEFS-Aerosol" and in Section 3.11 "Annual Budget". As for the difference of AGCM (Atmospheric General Circulation Model) in the two models, GEOS4-GOCART uses a dynamic core based on Lin and Rood (1996), and GEFS-Aerosols using the dynamic core FV3 (Finite Volume Scheme with Lagrangian Vertical Coordinate) was also developed based on the work of Lin and Rood (1996). Other configurations in AGCM, such as land models and microphysics, are quite different. Discussing their impact on aerosols is a very large topic that hopefully can be covered in future work.

Lin, S. J., & Rood, R. B. (1996). Multidimensional flux-form semi-Lagrangian transport schemes. Monthly Weather Review, 124(9), 2046-2070.

*-Line 310 'interannual variations' It is not clear why the authors attempt to analyze interannual variations with a 15-month data set.*

The authors sought to find an answer to the question "Can we use past emissions to predict future emissions, for example, for wildfire emissions?"

The 15-month data show no regularity in the nature sources of aerosol emissions on a global scale.

*-Line 332-333: 'The study of monthly and interannual variations in aerosol mass is important because it determines whether it is appropriate to use aerosol climatology fields rather than aerosol prognostic fields in weather forecasting to save computational resources.' I thought that the use of climatological, prescribed, or prognostic aerosols in the operational model is largely determined by the resource constraint. The study of monthly and interannual variations is important because it addresses many important aerosol-related scientific questions.*

Totally agree with the reviewer's point of view. For example, as NOAA/NCEP/EMC extend the global aerosol forecast from 5 days to 35 days, how to predict fire emissions in the 35-day forecast becomes more and more important.

---

## Editor Decision (ED1)

Dear Li Pan,

I had the opportunity to review the revised version of your manuscript as well as your replies to the referees' comments, and I appreciate the effort you put into addressing the comments and suggestions made by the referees.

However, after a thorough reevaluation of the revised manuscript, I regret to inform you that I believe the revisions made are not sufficient to address the concerns and issues raised by the referees. There are still several areas that require further attention and refinement.

Specifically, I would like to draw your attention to the following points that need further revision:

1. Requests for clarification or additional discussion raised by reviewers often indicate a lack of clarity of the manuscript or missing information, making it difficult for readers to follow the paper or argumentation. Therefore, additional explanations and clarifying statements should not only be provided in the author's response (e.g., RC1 comment on gravitational settling), but also be included in the revised version, except the authors consider a referee comment as inappropriate. In the latter case, a clear argument should be given in the author's response. Furthermore, a rephrasing of the initial text instead of a simple repetition often helps to increase clarity (e.g., RC1 comment on Fig. 6 and related discussion).

2. RC1 raised the point that the mass balance equation and associated processes did not mention the model's advection, diffusion, and physical processes. In your reply you wrote: "Although aerosols are affected by advection, diffusion, and physical processes, these processes are not specifically considered in the equations because they do not cause aerosols to leave/enter the system or change aerosol species." Unfortunately, this statement is not necessarily valid for a numerical model as the numerical schemes applied to solve advection, diffusion and other processes are not necessarily mass conserving, i.e., numerical artefacts might indeed lead to an artificial gain or loss of aerosol mass in the model, and this is exactly the referee's point. The same holds for the rather low model top. So to properly address this issue it needs to be shown and discussed how the model's advection, diffusion, physical processes and model top affect the aerosol mass balance, either by appropriate model experiments or by adding relevant references.

3. Both referees mentioned the lack of comparison / verification of your model results with observational data. As response to this point you added a short paragraph (lines 40-48) to the revised manuscript, listing a number of other studies which compared GEFS-aerosols with observational data. However, this paragraph still lacks a detailed discussion / summary of the outcome of the cited studies. Overall, the presentation of your data and results needs a more comprehensive discussion including results of relevant studies in the field.

4. RC1 suggested to show observations for sea salt near sea surface instead of AOD (Fig. 2). Such a figure has been included in the author's response to RC1, although without any discussion, but not in the revised version of the manuscript. Why? Please explain. Furthermore, I would suggest to add a figure showing the difference between GEFS and MERRA2. This would clearly facilitate the comparison.

5. RC1 suggested to show separate profile over land and sea in Fig. 10 and add some more discussion. I do not see any of this in the revised version, but also no related statement in the authors' response. Furthermore, you mentioned that for the evaluation of Fig. 11 the HIPPO experiment and CALIOP observations were used. However, it remains unclear to me how this evaluation has been done as I do not see any observational data plotted in Fig. 10 or 11.

6. In your response to RC1, last point related to L350-365, you provided the following paragraph, which is obviously copied from the model description section of your manuscript: "In GEFS-Aerosols: "The GOCART dry deposition protocol [Chin et al., 2000] was used for GEFS-Aerosols…." " How are these technical details related to the referee's comment? Please clarify and put your response into context. Furthermore, it would be beneficial to add a short statement to your paper that measurements of aerosol deposition fluxes are extremely challenging and therefore rather limited.

I understand that revising a manuscript can be a time-consuming process, and I sincerely appreciate your dedication to your research. However, to ensure that your work meets the high standards of our journal and contributes significantly to the field, I kindly request that you carefully address the issues mentioned above and **to revise your manuscript accordingly**. The list above is not exclusive, so please reconsider all points mentioned in the initial reviews, including RC2.

If you have any questions or need clarification on any of the points raised, please do not hesitate to contact me.

I look forward to receiving the next revised version of your manuscript.

Sincerely,

Andrea Stenke

---

## Author Response (AR3)

I am grateful to the editor for his comments that pointed out the shortcomings of this manuscript and gave me a second chance to improve it.

Requests for clarification or additional discussion raised by reviewers often indicate a lack of clarity of the manuscript or missing information, making it difficult for readers to follow the paper or argumentation. Therefore, additional explanations and clarifying statements should not only be provided in the author's response (e.g., RC1 comment on gravitational settling), but also be included in the revised version, except the authors consider a referee comment as inappropriate. In the latter case, a clear argument should be given in the author's response. Furthermore, a rephrasing of the initial text instead of a simple repetition often helps to increase clarity (e.g., RC1 comment on Fig. 6 and related discussion).

For RC1 comment on gravitational settling, the following discussion has been added to the manuscript.

- This scheme not only calculates the gravity settlement from the upper layer of the model to the lower layer of the model, but also calculates the gravity settlement from the bottom layer of the model to the ground. Meanwhile, the GOCART dry deposition protocol [Chin et al., 2000] was used for GEFS-Aerosols. Dry deposition in GEFS-aerosol was calculated by dry deposition velocity based on aerodynamic resistance, sublayer resistance and surface resistance. Therefore, aerosol gravity deposition and aerosol dry deposition are completely separated in GEFS-Aerosols.

For RC1 comment on Fig 6, this paragraph has been rewritten as:

- "To better understand the model error shown in Fig. 6, the GEFS-Aerosols output frequency was changed from every 3 hours (orange line in Fig. 7) to every hour (blue line in Fig. 7). Fig. 7 shows that the hourly variation in model error for dust simulations (in a 5-day simulation) is actually similar to that at 3-hour intervals, but the magnitude of peaks is reduced by about 60%, suggesting that model error is sensitive to model output frequency. "
- The linear assumption of aerosol deposition and emissions when testing the mass balance equation (Eq. 1) are the main cause of the model error. The deposition and emissions output by the GEFS-Aerosols diagnostic system are instantaneous values rather than cumulative values. Therefore, to calculate the cumulative amount of aerosol deposition or emissions over a model output time interval (e.g., three hours), simply multiply this value by three based on the linearity assumption. This treatment only affects deposition calculations for BC and OC (The daily emissions of BC and OC are constant.), but for dust and sea salt it affects not only deposition but also emissions calculations. At the same time, the wind threshold velocity makes the dust emissions more nonlinear than the source or sink terms of the other aerosol types. Therefore, the model errors for dust and sea salt are higher than those for BC and OC, while the model errors for dust are the highest. In general, when aerosol deposition or emissions increase, the error in calculating them in the analysis also increases due to linearity assumptions. For example, when aerosol emissions increase, in addition to BC and OC, the error in calculating aerosol emissions in the mass balance equation also increases. Correspondingly, the error in deposition calculations will also increase because an increase in ambient aerosol concentration will lead to an increase in the amount of deposition. This is why the model error for dust has a higher correlation with aerosol emissions than for BC and OC. However, for sea salt this correlation does not exist because sea salt emissions shown in Figure 6 are relatively stable. "

45
RC1 raised the point that the mass balance equation and associated processes did not mention the model's advection, diffusion, and physical processes. In your reply you wrote: "Although aerosols are affected by advection, diffusion, and physical processes, these processes are not specifically considered in the equations because they do not
50 cause aerosols to leave/enter the system or change aerosol species." Unfortunately, this statement is not necessarily valid for a numerical model as the numerical schemes applied to solve advection, diffusion and other processes are not necessarily mass conserving, i.e., numerical artefacts might indeed lead to an artificial gain or loss of aerosol mass in the model, and this is exactly the referee's point. The same holds for
55 the rather low model top. So to properly address this issue it needs to be shown and discussed how the model's advection, diffusion, physical processes and model top affect the aerosol mass balance, either by appropriate model experiments or by adding relevant references.

60 The following discussion has been added to the manuscript in response to comments from the reviewers and editor.
- "For a given system (such as the entire atmosphere), the amount of chemicals entering the system is equal to the amount of chemicals leaving the system."
- "Although aerosol mass is affected by advection, diffusion, and physical processes, these
65 processes are not specifically considered in the equations because they do not cause aerosols to leave/enter the system or change the aerosol species. However, these processes do change the concentration of aerosols in the atmosphere, and this effect is ultimately included in the "Initial", "Reaction", "Removal" and "Final" terms of the mass balance equation."
- "Theoretically, aerosol mass in GEFS-Aerosols simulations should be conserved, which means
70 that the model error should be zero if calculation accuracy is not taken into account. Possible reasons for the non-conservation of aerosol mass in GEFS-Aerosols as shown in Fig. 6 include: 1) The aerosol mass is not conserved in the advection, diffusion and physical processes of the model; 2) Aerosol leakage at the top of the model layer; 3) There are problems with calculating aerosol emissions and deposition in the mass balance equations; "
75 - "First, aerosol transport in GEFS-Aerosols is based on the FV3 dynamic core [Lin et al., 1994], which is also used in NASA-GOCART and GEOS-Chem. The mass conservation problem of this dynamical framework has been discussed by Lin and Rood [1996]. The physical processes of GEFS-aerosols are derived from the GFDL (Geophysical Fluid Dynamics Laboratory) cloud microphysics scheme [Lin et al., 1983], which strictly adheres to the conservation of moist
80 energy during phase changes. Secondly, the pressure at the top of the model in GEFS-Aerosols is set to 200 Pa. Since the pressure at the top of the model is low enough and the layers of the model are dense enough near the top [Campbell et al., 2022], the aerosol concentration in GEFS-Aerosols is the background concentration ($1 \times 10^{-16}$ µg/kg) in these layers. There may be mass conservation issues at the top of the model, but their impact is minimal."

85 Through the editor's comments, I realized there was a misunderstanding about how the mass balance equations were used in this study. Therefore I also added this discussion in the manuscript.

- "Because the linearity assumption in the analysis can lead to model errors as mentioned above, the model error shown in Fig. 6 does not represent the true model simulation error, but rather

90    the calculation error in the mass conservation analysis.  Therefore the main purpose of using the mass balance equation in this study is to verify the aerosol deposition and emissions calculated in the model budget analysis, rather than verifying whether aerosol mass is conserved in the GEFS-Aerosols."

Both referees mentioned the lack of comparison / verification of your model results with observational data. As response to this point you added a short paragraph (lines 40-48)
95  to the revised manuscript, listing a number of other studies which compared GEFS-aerosols with observational data. However, this paragraph still lacks a detailed discussion /summary of the outcome of the cited studies. Overall, the presentation of your data and results needs a more comprehensive discussion including results of relevant studies in the field

100

The following discussion has been added to the manuscript to summarize the work of Zhang et al. and Bhattacharjee et al.

- "These assessments found that GEFS-Aerosols captures not only major wildfire plumes in southern Africa, Siberia, the central Amazon, and central South America, as well as agricultural
105   fire plumes over India, but also high dust events in North Africa and the Arabian Peninsula; At the same time, GEFS-Aerosols has good performance in reproducing the seasonal variations at most surface observation sites dominated by dust and biomass plumes, as well as reproducing the vertical profiles of OC, BC, sulfate, dust and sea salt observed by ATOM. However, these findings are based on comparisons of AOD or aerosol concentrations and lack other assessments
110   beyond AOD and concentration."

RC1 suggested to show observations for sea salt near sea surface instead of AOD (Fig. 2). Such a figure has been included in the author's response to RC1, although without any discussion, but not in the revised version of the manuscript. Why? Please explain.
115  Furthermore, I would suggest to add a figure showing the difference between GEFS and MERRA2. This would clearly facilitate the comparison.

AOD represents aerosol column concentration and may better match the aerosol removal process. Because aerosol removal doesn't just come from the ground. AOD is derived from aerosol
120   concentration. One possibility is that the AODs may be very similar, but the aerosol concentrations are very different. I guess this is why the first reviewer asked how AOD is calculated in GEFS-Aerosols. A sea salt surface mass concentration plot has been added to Figure 2, and the following discussion is included in the manuscript:
- "Fig. 2 represents the monthly mean sea salt AOD (top) and surface mass concentration ($\mu g/m^3$)
125   (bottom) simulated in October 2019 from GEFS-Aerosols (left) and Modern-Era Retrospective analysis for Research and Applications Version 2 (MERRA-2) (right) [Molod et al., 2015]. GEFS-Aerosols and MERRA-2 show very similar results in simulating sea salt AOD, as does the distribution pattern of sea salt surface mass concentration. This is due to the fact that in GEFS-Aerosols, AOD is calculated using look-up tables (LUTs) of aerosol optical properties in the NASA
130   GOCART model, consistent with AOD calculations in MERRA2."

RC1 suggested to show separate profile over land and sea in Fig. 10 and add some more discussion. I do not see any of this in the revised version, but also no related statement in the authors' response. Furthermore, you mentioned that for the evaluation of Fig. 11 the HIPPO experiment and CALIOP observations were used. However, it remains unclear to me how this evaluation has been done as I do not see any observational data plotted in Fig. 10 or 11.

The following discussion has been added to the manusctipt

- "Fig. 11 is more representative of the vertical distribution of aerosols near the source (such as over land), while Fig. 10 is more representative of the vertical distribution of aerosols far away from the source (such as over the ocean)."

The difference in vertical distribution of aerosols over ocean and land has been discussed in the manuscript, for example Figure 10 "For aerosol species other than sea salt, aerosol mass peaks at pressure levels between 800 and 600 hPa."; for Figure 11, "The contribution of aerosol surface emissions to aerosol concentration decreases with increasing altitude".

The HIPPO experiments and CALIOP observations are only used for qualitative comparison with Figure 10. Because Figure 10 shows the simulation results in September 2019, later than the HIPPO experiment, and the CALIOP observations cited in our manuscript are earlier than our simulations. The following discussion has been added to the manuscript:

- "Note that the HIPPO experiment and CALIOP observations are only used for qualitative comparisons in this study because they are of different timing than the GEFS-aerosol simulations."

In your response to RC1, last point related to L350-365, you provided the following paragraph, which is obviously copied from the model description section of your manuscript: "In GEFS-Aerosols: "The GOCART dry deposition protocol [Chin et al., 2000] was used for GEFS-Aerosols…." " How are these technical details related to the referee's comment? Please clarify and put your response into context. Furthermore, it would be beneficial to add a short statement to your paper that measurements of aerosol deposition fluxes are extremely challenging and therefore rather limited.

I would say that the removal protocols used in GEFS-Aerosols may differ from those used in GOCART, but they still come from a standard protocol that is now widely accepted by the aerosol modeling community. The differences mainly arise from the way these schemes are parameterized in the model. Therefore, it is not surprising to see large differences. The lack of observations may give us a freedom to adjust the model at will. However, that doesn't mean it's correct.

The following discussion has been added to manuscript.

- "The budget analysis in Table 2 again demonstrates that two models can have completely different sources and sinks but end up with very similar concentration predictions, while at the same time it is difficult to discover which model is more correct. Because few observational are available for verification, especially for aerosol removal processes or net mass fluxes at surfaces. Measuring aerosol deposition fluxes is extremely challenging [Farmer et al., 2021], so such observations are rare. For example, the in-cloud mass scavenging efficiency of BC [Yang et al., 2019], the number of studies in this area is small but the reported data vary greatly, making it difficult to use for model evaluation."

- Author_Response-1
  -
  - The authors would like to thank the reviewers for volunteering their time to review this manuscript. Your comments make this manuscript better and better. I have carefully read your valuable suggestions, and the following is my reply.
  - ***This manuscript described a process-based budget analysis of the GEFS-Aerosols chemical transport model, including the processes of emissions, reactions and removal. This model budget analysis includes the comparison to the MERRA-2 and GEO4-GOCART, but has few verification with observations, making it hard to evaluate which process has big uncertainties.***
  - Bhattacharjee et al., 2023 (DOI: https://doi.org/10.1175/WAF-D-22-0083.1) evaluated the simulation results of the GEFS-Aerosols model using AOD data derived from satellite retrieval (MODIS and VIIRS), AOD data simulated by other models (MEERA2 and NGAC), and AOD data observed from 50 AERONET stations. The period of evaluation from August 2019 to August 2020 almost coincides with the time period of this study, namely from September 2019 to September 2020. In addition to the regular daily or monthly forecast evaluations of GEFS-Aerosols, three special events were also utilized to evaluate the performance of GEFS-Aerosols. These include dust events in Northwest Africa, agricultural fires in northern India and the August fire complex in northern California.
  - Zhang et al., 2022 (DOI: https://doi.org/10.5194/gmd-15-5337-2022) evaluated not only the AOD simulated by GEFS-Aerosols from 5 July to 30 November 2019, but also the aerosol concentrations simulated by GEFS-Aerosols during the 22-month ATOM (Atmospheric TOmography Mission) period from 2016 to 2019.
  - Those sentences has been added to lines 40 to 48 of the manuscript.
  - The following is the author's response to the detailed comment
  - ***Section 2.1 and 2.2. The mass balance equation and associated processes did not mention the model's advection, diffusion, and physical processes. How well has the aerosol mass been conserved in these processes? What's this model's top boundary treatment, and how does the model control the mass leakage through the domain top?***
  - The research object of mass balance equation is the total mass of aerosol in the atmosphere. Any aerosol mass fluxes (e.g. emissions and removals) leaving or entering the system are considered in the mass balance equation. The reaction is also included in the equation as it changes the aerosol species and the mass balance equation is calculated for the aerosol species. Although aerosols are affected by advection, diffusion, and physical processes, these processes are not specifically considered in the equations because they do not cause aerosols to leave/enter the system or change aerosol species. However, these processes do change the concentration of aerosols in the atmosphere, and this effect is ultimately reflected in the two terms "initial and final" in the mass balance equation.
  - The pressure at the top of the model in GEFS-Aerosols is 200 Pa. As shown in Figures 10 and 11, it is almost impossible for aerosols to be elevated to this level in

220 our simulations, and therefore, at this level, the aerosol concentration in GEFS-Aerosols is the background concentration (1 x 10$^{-16}$ µg/kg).

-

- ***The gravitational settling of aerosols is usually applied to mass movement from upper layers to the lowest layer of the model, which won't affect the total mass. The removal of aerosol mass from the model's lowest layer to ground***
225 ***surface refers to dry deposition, which should include the gravitational sediment. Does the GEFS-Aerosol's dry deposition scheme exclude the gravitational sedimentation? Please clarify.***

- The gravity scheme used in GEFS-Aerosol is mentioned in the manuscript lines 76 to 78 " the computation of gravitation settling for dust and sea salt are based on the
230 updated finite-difference scheme in WRF-CHEM (Ukhov et al, 2021)". This scheme not only calculates the gravity settlement from the upper layer of the model to the lower layer of the model, but also calculates the gravity settlement from the bottom layer of the model to the ground. Meanwhile, dry deposition in GEFS-aerosol was calculated by dry deposition velocity based on aerodynamic resistance, sublayer
235 resistance and surface resistance. Therefore, aerosol gravity deposition and aerosol dry deposition are completely separated in GEFS-Aerosols.

-

- ***Fig 2, and line 115, Are the sea salt AOD calculation method same in GEFS-Aerosol and MERRA-2? Besides AOD comparison, it is better to have mass***
240 ***concentration comparison with observations for sea salt near sea surface. How about the mass flux for other species?***

- In GEFS-Aerosols, AOD is calculated using a look-up table (LUT) of aerosol optical properties from the NASA GOCART model, so AOD calculations are consistent in GEFS-Aerosols and MERRA2.

[Figure]

Seasalt Surface mass conc. in mcg/m3 for Oct,2019

245 -

- The figure above shows a comparison of surface concentrations of sea salt aerosols in GEFS-Aerosols and MERRA2 in October 2019.

-

- ***Fig 3, 4 and line 117-120. Similar question, does the dry deposition of the***
250 ***lowest model layer exclude the sedimentation? It would better to show the net surface flux of sea salt in GEFS-Aerosol and MERRA-2, to show whether their emission and removal processes are balanced.***

- It can be seen from Table 2 that the total amount of sea salt emissions and removal processes in GEFS-Aerosols are basically balanced during the one year simulation period.
-
- ***Fig 6 and corresponding discussion around line 138-145. The model error for dust mass is highly correlated with the dust emission, but these correlations do not exist for BC, OC and sea salt. Any further discussion about the difference among these species, which processes result in the difference?***
- Indeed, the model errors for BC and OC are also highly correlated with emissions, as indicated in lines 151–152 of the updated manuscript. "The larger model errors for dust, OC and BC coincide with emission outbreaks for dust, OC and BC, which are shown as the emission changes (kg/s -> blue line) in Fig. 6".
- However, the model errors for dust are larger than those for BC, OC and sea salt. The reasons for this were discussed in lines 159 to 164 of the updated manuscript "The assumptions of linearity in aerosol deposition and emissions in the GEFS-Aerosols calculations are the main cause of its model error. This assumption only affects deposition calculations for BC and OC (The daily emissions of BC and OC are constant.), but for dust and sea salt it affects not only deposition but also emissions calculations. At the same time, the wind threshold velocity makes the dust emissions more nonlinear than the source or sink terms of the other aerosol types. Therefore, the model errors for dust and sea salt are higher than those for BC and OC, while the model errors for dust are the highest."
-
- ***Fig 10. What is the temporal and spatial extents of these profiles? Please clarify. It is better to separate the profiles over ocean and land, and have a deeper discussion for sea salt and land-source aerosols etc.***
- ***Line 282-290. It is more convincing if any direct comparison with observations can be included here, like CALIOP or in-situ measurements.***
- The data used in Fig. 10 are from September 2019 monthly averages and are global averages. Similar graphs were plotted for other months in this experiment. Since these figures are very similar, they are not shown in this manuscript.
- Both Fig. 10 and Fig. 11 represent the simulated vertical distribution of aerosols in GEFS-Aerosols. By comparing Fig. 10 and Fig. 11, we noticed that the different data processing methods when plotting Fig. 10 and Fig. 11 made the GEFS-aerosol vertical profiles in Fig. 10 and Fig. 11 very different. Fig. 11 shows that aerosol concentrations decrease with increasing altitude. However, Fig. 10 shows that at least for some aerosols (e.g. BC, OC, sulfate, and dust) concentrations are higher at higher altitudes. Even so, they are not contradictory. Because Fig. 11 (zonal distribution) is more representative of the vertical distribution of aerosols near the source area (e.g. land), while Fig. 10 is more representative of the vertical distribution of aerosols away from the source area (e.g. over the ocean). Therefore, to evaluate Fig. 11, the HIPPO experiment and CALIOP observations were used, since they measure vertical profiles of aerosols in remote regions.
-
- Added the following sentence into the manuscript: "The decrease in aerosol concentration with increasing altitude shown in Fig. 11 is significantly different from that shown in Fig. 10. Fig. 10 shows that certain aerosols (e.g. BC, OC, sulfate and dust) are more concentrated at higher altitudes. However, Fig. 10 and Fig. 11 are not

contradictory. Because Fig. 11 is more representative of the vertical distribution of aerosols near the source (such as over land), while Fig. 10 is more representative of the vertical distribution of aerosols far away from the source (such as over the ocean). Therefore, in the validation of Fig. 10, the HIPPO experiment and CALIOP observations were used because they measured the vertical profile of aerosols in remote areas."

-

- ***Table 2 and Section 3.11 It is better to make consistent by changing the units of wet/dry deposition and sediment from percent to Tg/Year, comparable to emission etc.***

-

- Three more columns were added to Table 2, indicating the amount of aerosol removed by wet deposition, dry deposition, and deposition (Tg/Y) in GEFS-aerosol, and sentence "and the total removal of aerosols (i.e. the sum of wet, dry and sedimentation) is almost equal to their total emissions (Table. 2)." was added to the text.

-

- ***Line 350-365. Giving the big discrepancies between the models, is there any observation available to verify aerosol removal process or surface net mass flux?***

-

- Measuring aerosol deposition fluxes is extremely challenging (Farmer et al., 2021), so observations are limited. For example, the in-cloud mass scavenging efficiency of BC (Yang et al., 2019), the number of studies in this area is small but the data reported vary widely, making them difficult to use for model evaluation.

-

- In GEFS-Aerosols: "The GOCART dry deposition protocol [Chin et al., 2000] was used for GEFS-Aerosols. Wet deposition in GEFS-Aerosols is the sum of large-scale wet removals and convective scavenging. The large-scale wet removal scheme is from WRF-CHEM

- (https://github.com/wrf-model/WRF/blob/master/chem/module_wetdep_ls.F), and the convective scavenge is calculated in FV3GFS physics and is based on the simplified Arakawa-Schubert (SAS) scheme [Pan et al, 1994; Zhang et al., 2022b]. The computation of gravitational settling for dust and sea salt are based on the updated finite-difference scheme in WRF-CHEM [Ukhov et al., 2021]."

-

- *Farmer, D. K., Boedicker, E. K., & DeBolt, H. M. (2021). Dry deposition of atmospheric aerosols: Approaches, observations, and mechanisms. Annual review of physical chemistry, 72, 375-397.*

- *Yang, Y., Fu, Y., Lin, Q., Jiang, F., Lian, X., Li, L., ... & Sheng, G. (2019). Recent advances in quantifying wet scavenging efficiency of black carbon aerosol. Atmosphere, 10(4), 175.*

-
-
-
-
-
-
-
-
-

- • Author_Respose-2

- • *My major comments:*
- • The authors would like to thank the reviewers for volunteering their time to review this manuscript. Your comments make this manuscript better and better. I have carefully read your valuable suggestions, and the following is my reply.
- • *-The authors plainly describe what they see, not what they learn from the analysis. The manuscript could be further improved if the authors present the results with more scientific insight.*
- • *-It is very difficult to verify simulated lifetime and annual emission and removal due to lack of observational evidence. The authors only compare the GEFS-Aerosols results with GEOS4-GOCART from Colarco et al. 2010. How about the AeroCom consensus?*
- • GEFS-Aerosols, a new global aerosol model developed by NOAA, became operational in September 2020, and the last study time of AeroCom Phase III was 2010, so GEFS-Aerosols did not have the opportunity to participate in AeroCom for inter-model comparison. However, GEFS-Aerosols has been participating in GAFIS (global air quality forecasting and information system) (https://community.wmo.int/en/activity-areas/gaw/science-for-services/gafis) since 2022, a WMO (World Meteorological Organization) organized project for models inter-comparison.
- • *Here are some minor comments for the authors to consider:*
- • *-Line 36 'As a first step towards this goal, 'GEFS-Aerosols was implemented to replace NGAC. The efforts to enable prognostic aerosol capability toward the goal started with the implementation of NGAC. It is not clear to me why the authors view the GEFS-Aerosols implementation as the "first step."*
- • NGAC is an offline model that was replaced in September 2020 by the online model GEFS-Aerosols. NCEP is continuing to develop the model, such as adding aerosol data assimilation to the system. The goal is to incorporate aerosol components into NOAA UFS (Unified Forecast System). As such, we see this replacement as the first step toward our ultimate goal.
- • *-Line 43-44: 'because these processes occur before the model output and they are the determinants of aerosol concentration." I agree that budget analysis is important to examine model's fidelity/performance. However, the justification "these processes occur before the model output and they are the determinants of aerosol concentration" is very odd and weak. Budget analysis can reveal whether the model have the bulk emission and removal processes right. Whether these tendency diagnostics are model output is totally irrelevant.*
- • I agree with the reviewer's comments. To clarify what I meant, the sentence has been changed to "because these processes are determinants of aerosol concentrations".
- • *-Line 40 'instead of focusing on aerosol concentration and aerosol optical depth (AOD) in a general aerosol evaluation'. Comparing GEFS-Aerosols model output with PM/AOD observations is needed to thoroughly assess the model performance and identify potential model deficit. It is certainly all right for the authors to focus on budget analysis in this manuscript. Since the*

*model vs observation evaluation has been conducted and reported in other papers [Lines 109-111], the authors should briefly describe the efforts.*

- "Bhattacharjee et al. [2023] evaluated the simulation results of the GEFS-Aerosols model using AOD data derived from satellite retrieval (MODIS and VIIRS), AOD data simulated by other models (MEERA2 and NGAC), and AOD data observed from 50 AERONET stations. The period of evaluation from August 2019 to August 2020 almost coincides with the time period of this study, namely from September 2019 to September 2020. In addition to the regular daily or monthly forecast evaluations of GEFS-Aerosols, three special events were also utilized to evaluate the performance of GEFS-Aerosols. These include dust events in Northwest Africa, agricultural fires in northern India and the August fire complex in northern California. Zhang et al., [2022b] evaluated not only the AOD simulated by GEFS-Aerosols from 5 July to 30 November 2019, but also the aerosol concentrations simulated by GEFS-Aerosols during the 22-month ATOM (Atmospheric TOmography Mission) period from 2016 to 2019." has been added to the text from lines 40 to 48.

- *-Line53 Eq1: Initial + Emissions + Reactions = Final + Removal*
- *Based on the governing equation, I'll probably present the equation as*
- *Final = Initial + Emissions + Reactions – Removal*
- The governing equation usually describe how a variable (such as aerosol concentration) changes when other variables change. Since this study focuses on the aerosol mass balance in GEFS-aerosols, the governing equation was converted to mass balance equation.

- *-Line 78: 2.3 GEFS-Aerosols.   Consider presenting this sub-section first in Section 2.*
- It has been changed
- *-Line 83: 'GOCART' Please define the acronym*
- Added at line 24
- *-Line 104: 'Fire Radiant Power (FRP) '.   Fire Radiative Power?*
- Corrected
- *-Line 113: 'These processes ultimately define the aerosol concentration and AOD output by the model.'   These processes ultimately determined 3-d aerosol distribution, which in term affect concentration and AOD. But this sentence is somehow odd.*
- It has been changed
- *-Line 115 'MERRA2'  MERRA-2 is also based on GOCART. Does sea salt emission and removal scheme in GEFS-Aerosols differ from those in MERRA2?*
- GEFS-Aerosols has the same sea salt emission mechanism as GEOS4-GOCART [Gong, 2003], but MERRA2 is based on GEOS5, which updated sea salt emission scheme [Randles et al., 2017]; the sea salt removal scheme in GEFS-Aerosols is also different from that in MERRA2.
- Gong, S. L.: A parameterization of sea-salt aerosol source function for sub-and super-micron particles, Global biogeochemical cycles, 17(4), Doi10.1029/2003GB002079, 2003.
- Randles, C. A., A. M., da Silva, V., Buchard, P. R., Colarco, A., Darmenov, R., Govindaraju, A., Smirnov, B., Holben, R., Ferrare, J., Hair, Y., Shinozuka and Flynn, C. J.: The MERRA-2 Aerosol

Reanalysis, 1980 Onward. Part 1: System Description and Data Assimilation Evaluation, Journal of Climate, 30(17), 6823-6850, 10.1175/jcli-d-16-0609.1, 2017

-
- *-Line 138 'Fig 6'   The principal behind the budget analysis is that aerosols net production is approximately equal to net loss when averaged over a long time (say multiple years).   It is not clear whether the monthly residual (Left side of Eq1 – Right side of Eq 1) should be interpreted as 'model error'.*
- Annual aerosol deposition and sedimentation of BC, OC, dust and sea salt are added to Table 2, and as the reviewer states, aerosol emissions are almost equal to their total removals. We assume that the difference between the left side of Equation 1 and the right side of Equation 1 should be zero. If not, it means there is an error inside the model or in our analysis.
- *-Line 155 "Therefore, the model errors for dust and sea salt are higher than those for BC and OC, while the model errors for dust are the highest.".   The text seems indicate that the model errors for dust and sea salt are caused by non-linearity in the emission/removal scheme.  This is not necessarily true.*
- I should say yes, it might be true. If we could calculate dust and sea salt emissions and removals more precisely in our analysis, the model error could be very close to zero and much smaller than the numbers we saw in Fig 6 and Fig 7.
- *-Line 159 'Global Aerosol Mass'.  It is insightful to specify when specific aerosol species reach max and min.  For instance, dust loading peaks in June and reached min in Nov.  This results are consistent with Africa dust activities.  However, it seems unnecessary for the authors to specify the exact date.*
- I agree and corrected. The total amount of dust is highly correlated with the intensity of dust activity in Africa.
- *-Line 172 'Annual trend'.  How annual trend can be inferred from one-year simulation?   Please clarify it.*
- Corrected to "In the simulated year, the trends for BC and OC masses are decreasing (16.4% and 22.3%, respectively) and the trends for dust and sea salt are increasing (24.9% and 16.0%, respectively); for sulfates the trend is almost constant with only a very slight decrease (8.09%)."

- *-Line 181-186.  The discussions about the partition can be presented in a table.*
- The partition of dust and sea salt emissions is shown in Table 1.
- *-Line 189 'Aerosol emissions are directly and indirectly related to their mass in the atmosphere'.   Aerosol loading is certainly related to their emissions, and aerosol emissions certainly affect aerosol mass.  However, the statement is very awkward.*
- The "indirect" mentioned in this sentence refers to sulfate, because there is no sulfate emission in GEFS-Aerosols. Sulfate is converted mainly from $SO_2$, which mainly comes from anthropogenic sources in GEFS-Aerosols.
- *-Line 224-225: 'the size distribution of aerosol emissions becomes too important for the removal process in GEFS-Aerosols simulations when the aerosol particle size is not changed in the model'  Please clarify this sentence.*
- For example, if the total dust emission is 50kg, of which dust1 emission is 5 kg, dust2 emission is 5 kg, dust3 emission is 10 kg, dust4 emission is 25 kg, dust4

emission is 5 kg, then finally 5 kg of dust is removed as dust1, 5 kg of dust is removed as dust2, 10 kg of dust is removed as dust3, 25 kg of dust is removed as dust4, and 5 kg of dust is removed as dust5. In summary, for each dust size, the amount emitted is the amount removed since the particle size of the dust does not change in the GEFS-Aerosols simulation.

- On the other hand, 100% OC is emitted as hydrophobic, but during the removal process, 50.5% OC is removed as hydrophobic and 49.5% OC is removed as hydrophilic, because hydrophobic OC can be converted into hydrophilic OC.

- ***-Line 229 'as they do not undergo a size (bin) change.' The GOCART is a bulk mass scheme. It's not clear to me why the authors expect bin change.***

- Please refer to the previous reply.

- ***-Line 265 'GOCART' Presume it's GOES4-GOCART. It does not hurt to make it clear.***

- Corrected

- ***-Line 263-269, The differences in lifetime between GEFS-Aerosols and GEO4-GOCART are attributed to model resolution and simulation period. Both model use GOCART scheme. The differences and similarities between the two GOCART schemes should be considered. The difference between the two host AGCMs should also be discussed. If identical emission and removal scheme are implemented in both GEFS and GEOS4, the emissions and removal fluxes from the two model will still be different. The model with more active moisture process may produce more wet removal. The model with more noisy wind field may produce higher dust emissions.***

- The similarities and differences between GEFS-Aerosol and GEOS4-GOCART in terms of aerosol deposition and emissions have been discussed in Section 2.1 "GEFS-Aerosol" and in Section 3.11 "Annual Budget". As for the difference of AGCM (Atmospheric General Circulation Model) in the two models, GEOS4-GOCART uses a dynamic core based on Lin and Rood (1996), and GEFS-Aerosols using the dynamic core FV3 (Finite Volume Scheme with Lagrangian Vertical Coordinate) was also developed based on the work of Lin and Rood (1996). Other configurations in AGCM, such as land models and microphysics, are quite different. Discussing their impact on aerosols is a very large topic that hopefully can be covered in future work.

- Lin, S. J., & Rood, R. B. (1996). Multidimensional flux-form semi-Lagrangian transport schemes. Monthly Weather Review, 124(9), 2046-2070.

- ***-Line 310 'interannual variations' It is not clear why the authors attempt to analyze interannual variations with a 15-month data set.***

- The authors sought to find an answer to the question "Can we use past emissions to predict future emissions, for example, for wildfire emissions?"

- The 15-month data show no regularity in the nature sources of aerosol emissions on a global scale.

- ***-Line 332-333: 'The study of monthly and interannual variations in aerosol mass is important because it determines whether it is appropriate to use aerosol climatology fields rather than aerosol prognostic fields in weather forecasting to save computational resources.' I thought that the use of climatological, prescribed, or prognostic aerosols in the operational model is largely determined by the resource constraint. The study of monthly and interannual variations is important because it addresses many important aerosol-related scientific questions.***

535
- Totally agree with the reviewer's point of view. For example, as NOAA/NCEP/EMC extend the global aerosol forecast from 5 days to 35 days, how to predict fire emissions in the 35-day forecast becomes more and more important.
-